

# The evaluation and enhancement of quality, environmental protection and safety in seaports

Danijela Tadic[1], Aleksandar Aleksic[1], Pavle Popovic[2], Slavko Arsovski[1], Ana Castelli[3], Danijela Joksimovic[3], Miladin Stefanovic[1]

[1]Faculty of Engineering, University of Kragujevac, Sestre Janjic 6, 34 000 Kragujevac, Serbia
[2]Luka Kotor a.d. Kotor, Park Slobode 1. 85330 Kotor, Montenegro
[3]Institute of Marine Biology, Dobrota bb - POBox 69, 85330 Kotor, Montenegro

*Correspondence to*: Pavle Popovic ( pavle.popovic@portofkotor.co.me )

**Abstract.** The evaluation and enhancement of business processes in any organization in an uncertain environment presents one of the main requirements of ISO 9000:2008, and has a key effect on competitive advantage and long-term sustainability. The aim of this paper can be defined as the identification and discussion of some of the most important business processes of seaports, and the performances of business processes and their key performance indicators (KPIs). The complexity and importance of the treated problem calls for analytic methods rather than intuitive decisions. The existing decision variables of the considered problem are described by linguistic expressions which are modelled by triangular fuzzy numbers (TFNs). In this paper, the modified fuzzy extended analytic hierarchy process (MFAHP) is proposed. The assessment of the relative importance of each pair of performances and their KPIs are stated as a fuzzy group decision making problem. By using MFAHP, the fuzzy rank of business processes of a seaport is obtained. The model is verified through an illustrative example with real life data, where the obtained data suggest measures which should enhance business strategy and improve KPIs. The future improvement is based on benchmark and knowledge sharing.

**Keywords:** process approach, seaport, fuzzy sets, analytic hierarchy process

## 1 Introduction

Changes in the business world, above all in the domain of politics, the economy and environment, demand the continuous improvement of business processes that are provided by strategic managers with the goal of increasing business in a seaport. A QMS conforming to ISO 9001:2008 should be considered as an important additional step, in terms of quality, because ISO 9001 also takes into account economic and financial aspects, design and development aspects, and introduces a management review for measurement and analysis of a process with the aim of improving performances (Poli et al., 2012). However, this important issue forces every organization to start either with ISO 9000 or TQM as a business strategy (Sedani and Lakhe, 2011). There are various ways in which an enterprise can claim that its QMS meets the requirements of ISO 9001 (Ali, 2014).



Lately, economics, geography and operations research including risk management (John et al., 2014) have dominated seaport research. The impact of multidisciplinarity and interdisciplinarity has significantly increased, and management studies have also substantially increased in the 2000s (Woo et al., 2011).

Motivation for this research comes from the fact that there are no research papers that treat seaports in the context of the process approach – an assessment of business processes' performances and their improvement which may be obtained by application of different methods. These methods are developed on a different mathematical and logical framework. According to results of good practice, it is known that it is almost impossible to enhance all business processes at the same time, having in mind overall complexity and definite resources – costs, time, human resources, etc. Enhancement activities are based on the already defined priority of business processes. Business processes may be described by different performances. Different preferences of business processes may vary depending on the needs of the seaport's business strategy, international standards related to seaport services, change of the environment, etc. According to the results of seaport good practice, the key performance indicators (KPIs) of business processes may be defined. It may be assumed that the defined performances and their KPIs do not have the same relative importance. In compliance with this assumption, it can be said that priorities of business processes can be stated as a multi-criteria optimization problem.

The wider objective of this research may be interpreted as an integration of the process approach, the management of business processes and multicriteria optimization methods. The mentioned integration includes: a) presentation of a seaport as a network of unrelated business processes so the overall success of the business processes may be assessed on the level of predefined criteria; b) the assessment of business processes by FAHP; c) definition of management initiatives which should lead to the improvement of business success; the order of taking management initiatives is based on the obtained rank of business processes.

This paper continues to investigate beyond the existing seaport literature and intends to consider all the significant performances, which have not had much attention in the management of business processes. The main contribution of the proposed model is that by its application, the fuzzy rank of business processes and the degree of belief that one business process is better than other with respect to each business' performance are obtained.

As the environment changes rapidly or becomes uncertain thus making the values of some performances and their relative importance difficult or impossible to quantify. All existing uncertainties can be adequately described by linguistic expressions which are modelled by applying the fuzzy sets theory (Klir and Folger, 1988; Zimmermann, 2001). The fuzzy sets theory resembles human reasoning in its use of approximate information and uncertainty to generate decisions. It may be suggested that the fuzzy approach to treating uncertainties in real-word applications has numerous advantages when compared to other approaches such as applying the probability theory, the rough set theory, etc. The appropriate technique for determining the rank of business processes with respect to all identified performances in a seaport is fuzzy AHP (FAHP). It is assumed that it is closer to human thinking, and that the relative importance of KPIs and performances of business processes are assigned according to a pairwise comparison matrix (Saaty, 1990). The issue of evaluation of the relative importance of performances and their KPIs may be based on the FAHP framework (Chan and Kumar, 2007; Kaya and



Kahraman, 2011; Tadic et al., 2015; Hsu, 2012; Kuo et al., 1999). It is assumed that all experts do not have equal importance, so in this paper, aggregation of the individual opinions of experts is performed by using Fuzzy Ordered Weighted Averaging (FOWA) (Merigo and Casanovas, 2008) (by analogy to Aleksic et al., 2013; Tadic et al., 2015).

The main difference between FAHP and the others in the literature is its modelling of a decision problem in a holistic

manner. This method offers a number of benefits: (1) the other multi-criteria methods experience difficulty in capturing uncertain and imprecise judgments of experts, (2) FAHP is an efficient tool for handling the fuzziness of the data involved in deciding the preferences or assessment of different decision variables. In this paper, a new approach for handling pair-wise comparison based on trapezoidal fuzzy numbers (Wu et al., 2004) is used.

In the literature, there are many developed approaches for handling FAHP. The use of the developed approach (Chang,

1996) does not involve cumbersome mathematical operation, and it has the ability to capture the vagueness of the human thinking style. Wang et al., (1996) have shown that the extent analysis method cannot estimate the true weights from a fuzzy comparison matrix and has led to quite a number of misapplications. There are many differences between traditional FAHP (Chang, 1996) and the FAHP which is proposed in (Wu et al., 2004). Firstly, fuzzy numbers can extend the range of a crisp comparison matrix of the AHP method. Secondly, in the proposed method, the weights of the criteria and preferences of an

alternative under each criterion are derived from the fuzzy preference rations, thus the developed approach allows a more reasonable description of the decision making process and reflects the thinking style of a human.

## 2 Materials and methods

Seaport research has been impacted from various perspectives, the economy, geography, management, and operational research (Woo et al., 2011). Their operations may be described with a lot of uncertainties, so lately there have been many

papers in literature that deal with risk management models (John et al., 2014) and metrics, proposed and numerically implemented to assess the overall performance of large systems, during natural disasters and their recovery – resilience (Shafieezadeh and Burden, 2014). This is due to the fact that much of the available data associated with port operations require a flexible but robust approach of handling as well as updating existing information with new data. As risk management activities are oriented to safety, port safety evaluation (Pak et al., 2015) is the first step in overall safety

enhancement. The common issue in the named papers that treat safety mangement is the need for approaches that may be used as an asset for determination of criteria for ranking of alternatives.

In the theory of strategic management, evaluation criteria of business processes are defined with respect to vision, mission and strategy of an organization (David, 2011). After quality management certification, determining of performances of business processes is based on pre-defined critical success factors (CSFs) (Oakland, 2004). Hutchins (2008) suggests that

performances of business processes can be defined with respect to the results of SWOT analyses and results of goal analysis. In practice, the determining of performances of business processes is performed by the owner of business processes of any organization with respect to the results of good practice and the defined management strategy.



## 2.1 Evaluation framework

The proposed evaluation procedure can be realized through steps which are presented in fig. 1 as follows.

Figure 1 – The evaluation procedure of seaport business processes by MFAHP

*Step 1.* The expert team needs to consist of the seaport owner, main manager, local government and the operational management of the seaport. Formally, this expert team is presented by a set of indices $\varepsilon = \{1, ..., e, ..., E\}$. The index for an expert is denoted as e, and E is the total number of experts. The members of the expert team have different influence in the considered decision making process. The importance of experts, $w_e$ , e=1,..,E should be determined with respect to the results of good practice.

*Step 2.* The identified performances can be presented by the set of indices $\kappa = \{1, ..., k, ..., K\}$ . The index for a performance is denoted as k, k=1,..,K and K is the total number of identified performances. Each performance k, k=1,..,K is decomposed into KPIs. Generally, KPIs under performance k, k=1,..,K are presented by the set of indices $\varphi_k = \{1, ..., j, ..., J_k\}$ . The total number of KPIs under performance k, k=1,..,K is denoted as $J_k$ , and j is the index for KPI j, j=1,.., $J_k$. The assessment of the relative importance of performances and their KPIs is performed by an expert team (by analogy to AHP).

*Step 3.* The reference model of an organization (in this case a seaport) may be seen as a general model which can be used for gaining other forms of models (Spiegel and Caulliraux, 2012). In compliance with this, an organization may be viewed as a network of interrelated processes that are focused towards achieving organizational goals (Oakland, 2004). The defining of seaport business processes is based on the process approach (ISO 9000:2008), and assessment of seaport operational management (quality manager, environmental manager and security manager). The identified business processes are presented by the set of indices $\iota=\{1, ..., i, ..., I\}$. The total number of treated business processes is I and i, i=1,..,I is the index of the business process. The assessment of the relative preference value of each pair of business processes is achieved by operational management. They make a decision by consensus.

*Step 4.* Experts and operational managers use the pre-defined linguistic expressions, which are modelled by TFNs. The shape of the membership functions can be obtained based on one's experience, the subjective belief of decision makers, and their knowledge. Jointly used shapes of triangular function offer a good compromise between descriptive power and computational simplicity. For this purpose, the five linguistic expressions are proposed, and they are modelled by TFNs as follows:

*very low importance/preferency*: VL = (x; 1,1,2)

*low importance/preferency*: L = (x; 1,2,3)

*moderate importance/preferency*: M = (x; 2,3,4)

*high importance/preferency*: H = (x; 3,4,5)

*very high importance/preferency*: VH= (x; 4,5,5)

The domains of these TFNs are defined into interval [1-5]. Value 1, and value 0 denote that one performance or KPI is as important, or unimportant, as any identified performances or KPIs under each treated performance.



*Step 5.* The aggregation of individual opinions into a group consensus is calculated by the performed Fuzzy Ordered Weighted Aggregation (FOWA) operator (Merigo and Casanovas, 2008).

*Step 6.* The weights vector of performances and weights vector of KPIs under each performance and the preference vector of business processes with respect to each KPI are determined by FAHP which is developed in (Wu et al., 2004).

*Step 7.* The overall preference index of each business process with respect to all performances and their weights is presented by fuzzy sets obtained using the standard operations of fuzzy sets theory (Klir and Folger, 1988; Zimmermann, 2001). The overall preference index of each business process is described by a TFN.

*Step 8.* The ranking of business processes is performed according to the overall index of preference. The rank of business processes corresponds to the rank of TFNs which are described by overall indices' preferences. The ranking of the TFNs $\tilde{p}_i$ , i=1,..,I and the calculating of the degree of belief that other business processes can be better than the business process which is placed in first place in the rank are based on a method for comparison of fuzzy numbers (Bass and Kwakernaak, 1977; Dubois and Prade, 1979).

## 2.2 Notation

In the course of easier understanding of the proposed Algorithm, in this Section the notation is given.

- $\widetilde{W}_{kk'}^e = \left(x; l_{kk'}^e, m_{kk'}^e, u_{kk'}^e\right)$ – is the TFN describing the relative importance of performance k over performance k', $k, k' = 1, .., K; k \neq k'$ which is given by expert e, e=1,..,E; the lower and upper bounds of TFN $\widetilde{W}_{kk'}^e$ are denoted as $l_{kk'}^e, u_{kk'}^e$, respectively and modal value $m_{kk'}^e$

- $W_{kk'}$ - is the TFN describing the aggregated relative importance of performance k over performance k', $k, k' = 1, .., K; k \neq k'$

- $\widetilde{W}_{jj'}^e = \left(x; l_{jj'}^e, m_{jj'}^e, u_{jj'}^e\right)$ – is the TFN describing the relative importance of KPI j over KPI j', under performance k, $j, j' = 1, .., J_k; j \neq j'; k = 1, .., K$; the lower and upper bounds of TFN $\widetilde{W}_{jj'}^e$ are denoted as $l_{jj'}^e, u_{jj'}^e$, respectively and modal value $m_{jj'}^e$

- $W_{jj'}$ - is the TFN describing the aggregated relative importance of KPI j over KPI j', under performance k, $j, j' = 1, .., J_k; j \neq j'; k = 1, .., K$

- $\widetilde{P}_{ii'} = \left(x; l_{ii'}, m_{ii'}, u_{ii'}\right)$ - is the TFN describing the relative preference of business process i over business process i', $i, i' = 1, .., I; i \neq i'$ under each KPI j, j=1,..,J$_k$; k = 1, .., K; the lower and upper bounds of TFN $\widetilde{P}_{ii'}$ are denoted as $l_{ii'}, u_{ii'}$, respectively and modal value $m_{ii'}$

- $\tilde{w}_k = \left(x; l_k, m_k, u_k\right)$ is the TFN describing the weight of performance k, k=1,..,K

- $\tilde{w}_j^k = \left(x; l_j^k, m_j^k, u_j^k\right)$ is the TFN describing the weight of KPI j under performance k, j=1,..,J$_k$; k = 1, .., K



- $\tilde{p}_{ij}^{\,k} = \left(y; l_{ij}^{k}, m_{ij}^{k}, u_{ij}^{k}\right)$ is the TFN describing the preference of business process i, i=1,..,I under KPI j=1,..,$J_k$ of performance k, k=1,..,K

- $\tilde{a}_{i}^{\,k} = \left(y; l_{i}^{k}, m_{i}^{k}, u_{i}^{k}\right)$ - is the TFN describing the preference index of business process i, i=1,..,I under performance k, k=1,..,K

$\tilde{a}_i = (x; L_i, M_i, U_i)$- is the TFN describing the overall preference index of business process i, i=1,..,I with respect to all identified performances and their weights.

**2.3 The proposed Algorithm**

The proposed procedure can be realized through steps.

*Step 1.* The fuzzy rating of the relative importance of each pair of performances and their KPIs are described by each expert and presented by TFN $\widetilde{W}_{kk'}^{e} = \left(x; l_{kk'}^{e}, m_{kk'}^{e}, u_{kk'}^{e}\right)$, k=1,..,K, and $\widetilde{W}_{jj'}^{e} = \left(x; l_{jj'}^{e}, m_{jj'}^{e}, u_{jj'}^{e}\right)$, j=1,..,$J_k$. The aggregated value of the considered variables are given by using FOWA (Eq. (1)):

$$\widetilde{W}_{kk'} = \left(x; l_{kk'}, m_{kk'}, u_{kk'}\right) = \sum_{e=1}^{E} w_e \cdot \widetilde{W}_{kk'}^{e}, k = 1,..,K; e = 1,...,E \qquad \text{Eq. (1)}$$

Similarly, the aggregated value of the relative importance of each pair of KPIs under the identified performance is determined.

*Step 2.* The weights vector of performances, weights vector of KPIs and preference vector of business processes under identified KPIs are calculated by applying the modified procedure which is developed in (Wu et al., 2004). The developed procedure is illustrated on the example of determination of the performances' weights vector in compliance with Eq. (2) and Eq. (3).

Order

$$\alpha_k = \left[\prod_{k=1}^{K} l_{kk'}\right]^{1/K}, \quad \beta_k = \left[\prod_{k=1}^{K} m_{kk'}\right]^{1/K}, \text{ and } \chi_k = \left[\prod_{k=1}^{K} u_{kk'}\right]^{1/K}, k = 1,..,K \qquad \text{Eq. (2)}$$

and

$$\alpha = \sum_{k=1}^{K} \alpha_k, \quad \beta = \sum_{k=1}^{K} \beta_k \text{ and } \chi = \sum_{k=1}^{K} \chi_k, \quad k = 1,..,K$$

Then the weight of performance, k=1,..,K is calculated as:

$$\tilde{w}_k = \left(\alpha_k \cdot \chi^{-1}, \beta_k \cdot \beta^{-1}, \chi_k \cdot \alpha^{-1}\right) = \left(y: l_k, m_k, u_k\right), k = 1,...,K \qquad \text{Eq. (3)}$$





In a similar way (Eq. (2) and Eq. (3)), the weight of KPI j, $\widetilde{w}_j^k = \left( y; l_j^k, m_j^k, u_j^k \right)$, j=1,...,$J_k$; k = 1,..,K and preference of

business process i $\widetilde{p}_{ij}^k = \left( y; l_i^j, m_i^j, u_i^j \right)$, i=1,...,I are calculated.

*Step 3.* The preference index of business process i, i=1,..,I under performance k can be calculated as (Eq. (4)):

$$\widetilde{a}_i^k = \sum_{j=1}^{J_k} \widetilde{w}_j^k \cdot \widetilde{p}_{ij}^k, \ i=1,...,I; \ j=1,...,J_k; \ k = 1,..,K \qquad \text{Eq. (4)}$$

*Step 4.* The overall preference index of business process i, i=1,..,I can be calculated as (Eq. (5)):

$$\widetilde{a}_i = \sum_{k=1}^{K} \widetilde{w}_k \cdot \sum_{j=1}^{J_k} \widetilde{w}_j^k \cdot \widetilde{p}_{ij}^k, \ i=1,...,I; \ j=1,...,J_k; \ k = 1,..,K \qquad \text{Eq. (5)}$$

*Step 5.* Rank all $\widetilde{a}_i$ in the decreasing order of $M_i, i = 1,..,I$.

*Step 6.* Calculate degrees of belief that any $\widetilde{a}_i$, i=1,...,I, $i \neq i^*$, is higher than $\widetilde{a}_{i^*}$, using the method given in (Bass and Kwakernaak, 1977; Dubois and Prade, 1979).

**3 Analysis of performances, KPIs and business processes in a seaport**

The product of seaports belongs to generic product categories called service (ISO 2000:2007). Respecting ISO 20000-1:2010 (point 2.15) and the above definition of the service term, it can be said that seaports can be denoted as service providers. The management of services to meet business requirements (ISO 20000-1:2010) can be maintained, amongst all, by application of a continual improvement principle of business processes. The ranking of business processes is stated as a problem which

has three levels of hierarchy, and the performances, KPIs of performances and the business processes will be further discussed. The considered performances of business processes are:

- Quality,
- Environmental protection,
- Seaport safety.

These performances can be decomposed into various other KPIs which are described below.

*Quality.* Quality is defined as the degree to which a set of inherent characteristics fulfil requirements (ISO 9000:2005) so fuzzy sets may be used when it needs to be assessed (Yaqiong et al., 2011). This performance of a business process has a high impact on customers, income and indirectly on long-term sustainability of the seaport. The KPIs affecting this performance can be determined with respect to literature data and as a result of good practice. KPIs of the quality (Tadic et

al., 2013) of seaport services are derived from ISO 9001:2008 and Resolution 10011 and can be stated as follows:




*Quality of the seaport services.* A seaport usually defines this KPI through the satisfaction and loyalty (ISO 10002:2014; ISO 10003:2007) of customers. It is supported by quality of internal customer oriented activities of the seaport and customers' perception of these activities.

*Average number of customers.* This KPI is very important for overall profit, local community and company image. The
impact on the local community is important since customers satisfy their needs in a seaport and by using the infrastructure around it (hotel services, banking services, shops, etc.).

*Average number of vessels in the queue.* As a seaport is customer oriented, this number should be as low as possible so the satisfaction of vessel owners and passengers will be increased. Also, this performance is important for different organizational units of the seaport such as repair services or services for loading and unloading vessels. It should be assessed
in communication with different services in the seaport that define approach positions of vessels and anchoring places.

*Pilotage operation of the vessel.* This performance is important from the perspective of vessel owners, customers and seaport management. All of them always require the minimum time needed for placing vessels in the limited seaport infrastructure. This should lead to overall cost minimization.

*Environmental protection.* In seaports worldwide, many accidents may occur leading to pollution of the environment and
biodiversity change. This is further propagated to the decreasing of business effectiveness in a seaport and in the worst case scenario, it can lead to total stoppage of provision of seaport services. It is important that maritime transport operates in a safe, secure and environmentally friendly way, so the EU has engaged legislation under port state control Directive 95/21. Besides this, ISO 14001:2004 sets out the criteria for an Environmental Management System (EMS) so in compliance with its demands, KPIs that describe environmental protection can be measured in terms of the following:

*Quality of air.* The level of air quality is important from the perspective of public health and change of biodiversity. It is defined and should be assessed through the level of smoke, dust and harmful gasses present. According to the evidence data, around 95.75 % $CO_2$ is emitted in the air, 22 % nitrogen oxides, 0.6 % sulfur oxides, etc. all of these could lead to the greenhouse effect and to damage of the ozone layer.

*Water quality.* This KPI is related to protection of sea biodiversity, tourists and the local community. The level of water
quality depends on the presence of micro biological, mechanical and chemical substances which are discharged by vessels entering the seaport.

*Noise.* This is significant from the perspective of customers. The other interested party is management of a seaport since noise represents a source of pollution. The increase in noise level may lead to change in biodiversity and to the minimization of profit since it reduces the satisfaction of customers and other stakeholders.

*Hazardous substances.* Hazardous substances may be generated in the majority of technical processes in a seaport and they potentially represent the most dangerous pollution sources for the environment. The management of hazardous substances is a very important task of a seaport having in mind biodiversity, public health and long-term sustainability.

*Seaport safety.* This significant performance has to meet legislative demands and it has a serious impact on seaport competitiveness. Different accidents could occur in ports causing extensive loss of lives, damage to vessels and cargo, and





serious water pollution and changes in biodiversity. Based on the literature review, the KPIs of seaport safety may be defined (Pak et al., 2015; Trbojevic and Carr, 2000). Based on the literature review (Pak et al., 2015) and evidence data of Montenegro seaports, the following KPIs are identified as the most significant:

*Vessel safety*. This KPI is related to the number of accidents caused by the collision of vessels in port, collision of vessels in
the port docks, unmooring from the dock, small boats capsizing, etc. In recent years, the safety of vessels in port has also been affected by the ability to hijack ships. This KPI may be assessed taking into account (Trbojevic and Carr, 2000) size, type, age, crew, maneuverability, pilotage requirements and escourting requirements.

*Traffic volume*. A traffic-related factor may be seen as 'Volume of traffic inside a port' (Pak et al., 2015). While assessing this KPI, a comprehensive database of port accidents may be used.

*Weather sea condition and channel condition*. This KPI may be addressed to: (1) weather conditions, such as wind speed, sea state and visibility (Balmat et al., 2009) and (2) channel conditions including the perspectives of depth, complexity, and width (Pak et al., 2015).

*Other safety factors*. Many factors impact safety of the port so they may be addressed as one joint KPI and they should be taken into account. These factors are fire safety, communication in port, terrorist attacks, natural disasters, etc.

## 15  4 Application of FAHP in business processes' ranking

The proposed model is tested on Kotor seaport located in a region which is protected under national legislation. In recent years, the seaport has been certified with ISO 9001:2008 and ISO 14001:2004. This seaport is a relatively small port so this fact is taken into account during the definition of a reference model of the organization.

In literature from business process management, processes of seaport services represent the processes of realization
(Arsovski, 2013). The number and type of business processes in a seaport is defined with respect to APQC and process owner opinion. A short description of the selected business processes in ports is further discussed.

*Planning and service monitoring* (p=1). It covers a set of activities to be implemented under the common goal of the process (responsibility for each activity, resources, timelines, and desired outputs from each activity in terms of the characteristics of services and processes). This process corresponds to the process Plan for and align supply chain resources which is defined
in APQC specification.

*Technology management of service providing* (p=2). It covers standard procedures for access of the vessels to the port, vessel pilotage procedures, maintenance procedures of vessels, port transportation, disembarking procedures, and procedures for cleaning, etc.

*Maintenance of infrastructure* (p=3). It covers maintenance procedures of docks, cranes, as well as other transport
manipulating systems, warehouses, roads, etc. This process corresponds to the process of Manage Logistics and Warehousing (respecting APQC).


*Management of Environmental Health and Safety* (p=4). It is defined in compliance with APQC specification and it is important from the perspective of seaport sustainability. The effectiveness of this business process is important for the management of the port and the local and state administration.

*Business activities in seaport* (p=5). This is a complex business process where a lot of different activities are defined and realized according to APQC and literature data (Medison, 2005). These activities are: material purchase, service delivery to seaport customers, marketing and service sale, management of customer demands, management of information technology and knowledge, management of financial resources and management of external relations.

The fuzzy-pair wise comparison matrix of the relative importance of evaluation criteria is presented (according to Step 1 of the proposed Algorithm):

$$\begin{bmatrix} (x;1,1,1) & M,H,(x;1,1,1),L & 1/L,1/VL,1/L,(x;1,1,1) \\ 1/M,1/H,(x;1,1,1),1/L & (x;1,1,1) & 1/M,1/H,(x;1,1,1),1/VL \\ L,(x;1,1,1),L,(x;1,1,1) & M,H,(x;1,1,1),VL & (x;1,1,1) \end{bmatrix}$$

Application of FOWA is illustrated by the following example. The aggregated relative importance of quality performance (k=1) over environmental protection performance (k=2) can be calculated as:

$$\widetilde{W}_{12} = 0.4 \cdot (x;2,3,4) + 0.3 \cdot (x;3,4,5) + 0.2 \cdot (x;1,1,1) + 0.1 \cdot (x;1,2,3) = (x;2,2.8,3.6)$$

The fuzzy pair-wise comparison matrix of the aggregated relative importance of performances is:

$$\begin{bmatrix} (x;1,1,1) & (x;2,2.8,3.6) & (x;0.45,0.7,1) \\ (x;0.28,0.36,0.5) & (x;1,1,1) & (x;0.41,0.51,0.58) \\ (x;1,1.43,2.22) & (x;1.72,1.96,2.44) & (x;1,1,1) \end{bmatrix}$$

The procedure for calculating quality weight is presented as follows (Step 2 of the proposed Algorithm):

$$\alpha_1 = \left[ \prod_{k=1}^{3} 1 \cdot 2 \cdot 0.35 \right]^{1/3} = 0.89, \quad \beta_1 = \left[ \prod_{k=1}^{3} 1 \cdot 2.8 \cdot 0.7 \right]^{1/3} = 1.25, \text{ and } \chi_1 = \left[ \prod_{k=1}^{3} 1 \cdot 3.6 \cdot 1 \right]^{1/3} = 1.53$$

and

$$\alpha = \sum_{k=1}^{K} \alpha_k = 2.65, \quad \beta = \sum_{k=1}^{K} \beta_k = 3.22, \text{ and } \chi = \sum_{k=1}^{K} \chi_k, = 3.95$$

Then the weight of quality performance (k=1) is calculated in compliance with Eq. (2) and Eq. (3):

$$\widetilde{w}_1 = \left( 0.89 \cdot 3.95^{-1}, 1.25 \cdot 3.22^{-1}, 1.53 \cdot 2.65^{-1} \right) = (y;0.24,0.39,0.58)$$

Similarly, weights of the rest of the performances are calculated:





$\tilde{w}_2 = (y;0.12,0.17,0.25)$, and $\tilde{w}_3 = (y;0.30,0.44,0.66)$.

The fuzzy pair wise comparison matrix of the KPIs under quality performance is:

$$\begin{bmatrix} (x;1,1,1) & (x;10.78,0.95,1) & (x;1.25,2,2.70) & (x;0.85,1,1) \\ (x;1,1.05,1.28) & (x;1,1,1) & (x;1.40,210,2.90) & (x;0.50,1,1) \\ (x;0.37,050,0.80) & (x;0.34,0.48,0.71) & (x;1,1,1) & (x;0.34,0.55,0.87) \\ (x;1,1,1.18) & (x;1,1,2) & (x;1.15,1.82,2.94) & (x;1,1,1) \end{bmatrix}$$

By using the procedure developed in (Wu et al., 2004), the weights of sub criteria under quality performance are:

$\tilde{w}_1^1 = (0.19,0.28,0.38)$, $\tilde{w}_2^1 = (0.19,0.29,0.41)$, $\tilde{w}_3^1 = (0.09,0.14,0.25)$, and $\tilde{w}_4^1 = (0.21,0.28,0.41)$.

The fuzzy pair wise comparison matrix of the KPIs under environmental protection performance is:

$$\begin{bmatrix} (x;1,1,1) & (x;0.22,0.28,0.40) & (x;0.31,0.48,0.80) & (x;1,1,1.30) \\ (x;2.50,3.57,4.55) & (x;1,1,1) & (x;1.40,2.10,2.90) & (x;0.50,1,1) \\ (x;1.25,2.08,3.23) & (x;0.34,0.48,0.71) & (x;1,1,1) & (x;0.34,0.55,0.85) \\ (x;0.87,1,1) & (x;1,1,2) & (x;1.18,1.82,2.94) & (x;1,1,1) \end{bmatrix}$$

The weights of KPIs under environmental protection performance are:

$\tilde{w}_1^2 = (0.09,0.14,0.25)$, $\tilde{w}_2^2 = (0.21,0.39,0.59)$, $\tilde{w}_3^2 = (0.11,0.20,0.36)$, and $\tilde{w}_4^2 = (0.18,0.27,0.48)$

The fuzzy pair wise comparison matrix of the KPIs under safety criterion is:

$$\begin{bmatrix} (x;1,1,1) & (x;0.67,0.90,1) & (x;2.30,3.30,4.30) & (x;3.50,4.50,5) \\ (x;1,1.11,1.49) & (x;1,1,1) & (x;2.30,3.30,4.30) & (x;3.80,4.80,4.90) \\ (x;0.23,0.3,0.43) & (x;0.23,0.3,0.43) & (x;1,1,1) & (x;1.50,2.50,3.50) \\ (x;0.20,0.22,0.29) & (x;0.20,0.21,0.26) & (x;0.29,040,0.67) & (x;1,1,1) \end{bmatrix}$$

The weights of KPIs under safety performance are:

$\tilde{w}_1^3 = (0.26,0.38,0.52)$, $\tilde{w}_2^3 = (0.29,0.41,0.58)$, $\tilde{w}_3^3 = (0.09,0.14,0.22)$, and $\tilde{w}_4^2 = (0.06,0.07,0.11)$.

Similarly, the fuzzy pair-wise comparison matrices of the business processes' preference are presented.

20 *Quality performance*

    *a) Quality of the seaport services.*





$$\begin{bmatrix} (x;1,1,1) & L & L & H & 1/VH \\ 1/L & (x;1,1,1) & M & VL & 1/VH \\ 1/L & 1/M & (x;1,1,1) & 1/L & 1/H \\ 1/H & 1/VL & L & (x;1,1,1) & 1/L \\ VH & VH & H & L & (x;1,1,1) \end{bmatrix}$$

$\tilde{p}_{11}^1 = (0.12, 0.21, 0.36)$, $\tilde{p}_{21}^1 = (0.06, 0.08, 0.17)$, $\tilde{p}_{31}^1 = (0.07, 0.12, 0.23)$, $\tilde{p}_{41}^1 = (0.07, 0.12, 0.22)$ and $\tilde{p}_{51}^1 = (0.28, 0.47, 0.72)$.

b)  *Average number of customers*

$$\begin{bmatrix} (x;1,1,1) & VH & H & M & H \\ 1/VH & (x;1,1,1) & VL & VH & 1/H \\ 1/H & 1/VL & (x;1,1,1) & 1/H & 1/VH \\ 1/M & 1/VH & H & (x;1,1,1) & 1/L \\ 1/H & H & VH & L & (x;1,1,1) \end{bmatrix}$$

5  $\tilde{p}_{12}^1 = (0.28, 0.43, 0.62)$  ,  $\tilde{p}_{22}^1 = (0.14, 0.19, 0.29)$  ,  $\tilde{p}_{32}^1 = (0.04, 0.06, 0.09)$  ,  $\tilde{p}_{42}^1 = (0.07, 0.10, 0.16)$  and

$\tilde{p}_{52}^1 = (0.14, 0.23, 0.34)$.

c)  *Average number of vessels in the queue*

$$\begin{bmatrix} (x;1,1,1) & VH & H & L & M \\ 1/VH & (x;1,1,1) & L & M & VL \\ 1/H & 1/L & (x;1,1,1) & L & (x;1,1,1) \\ 1/L & 1/M & 1/L & (x;1,1,1) & 1/VH \\ 1/M & 1/VL & (x;1,1,1) & VH & (x;1,1,1) \end{bmatrix}$$

10  $\tilde{p}_{13}^1 = (0.26, 0.44, 0.69)$, $\tilde{p}_{23}^1 = (0.11, 0.17, 0.32)$, $\tilde{p}_{33}^1 = (0.08, 0.13, 0.22)$, $\tilde{p}_{43}^1 = (0.05, 0.07, 0.14)$ and $\tilde{p}_{53}^1 = (0.12, 0.17, 0.27)$

.

d)  *Pilotage operation of the vessel*

$$\begin{bmatrix} (x;1,1,1) & L & VL & (x;1,1,1) & 1/VH \\ 1/L & (x;1,1,1) & M & VH & 1/L \\ 1/VL & 1/M & (x;1,1,1) & M & 1/VH \\ (x;1,1,1) & 1/VH & 1/M & (x;1,1,1) & 1/M \\ VH & L & VH & M & (x;1,1,1) \end{bmatrix}.$$

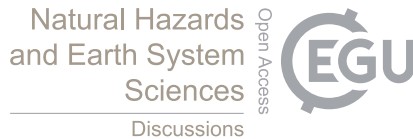

$$\tilde{p}_{14}^1 = (0.10, 0.14, 0.23) \quad , \quad \tilde{p}_{24}^1 = (0.13, 0.22, 0.39) \quad , \quad \tilde{p}_{34}^1 = (0.07, 0.12, 0.19) \quad , \quad \tilde{p}_{44}^1 = (0.06, 0.08, 0.12) \quad \text{and}$$

$$\tilde{p}_{54}^1 = (0.27, 0.45, 0.67).$$

*Environmental protection*

5  a) *Quality of air*

$$
\begin{bmatrix}
(x;1,1,1) & 1/L & 1/L & M & (x;1,1,1) \\
L & (x;1,1,1) & (x;1,1,1) & M & (x;1,1,1) \\
L & (x;1,1,1) & (x;1,1,1) & L & VL \\
1/M & 1/M & 1/L & (x;1,1,1) & L \\
(x;1,1,1) & (x;1,1,1) & 1/VL & 1/L & (x;1,1,1)
\end{bmatrix}
$$

$$\tilde{p}_{11}^2 = (0.11, 0.18, 0.32), \ \tilde{p}_{21}^2 = (0.17, 0.27, 0.40), \ \tilde{p}_{31}^2 = (0.15, 0.25, 0.44), \ \tilde{p}_{41}^2 = (0.07, 0.12, 0.23) \text{ and } \tilde{p}_{51}^2 = (0.10, 0.17, 0.25).$$

  b) *Water quality and c) Noise*

$$
\begin{bmatrix}
(x;1,1,1) & 1/M & 1/M & H & (x;1,1,1) \\
M & (x;1,1,1) & VL & H & VL \\
M & 1/VL & (x;1,1,1) & M & L \\
1/H & 1/H & 1/M & (x;1,1,1) & M \\
(x;1,1,1) & 1/VL & 1/L & 1/M & (x;1,1,1)
\end{bmatrix}
$$

$$\tilde{p}_{12}^2 = \tilde{p}_{13}^2 = (0.10, 0.15, 0.24) \ , \ \tilde{p}_{22}^2 = \tilde{p}_{23}^2 = (0.20, 0.30, 0.56) \ , \ \tilde{p}_{32}^2 = \tilde{p}_{32}^2 = (0.16, 0.32, 0.51) \ , \ \tilde{p}_{42}^2 = \tilde{p}_{43}^2 = (0.06, 0.10, 0.17)$$

$$\text{and } \tilde{p}_{52}^2 = \tilde{p}_{53}^2 = (0.07, 0.13, 0.20).$$

15  d) *Hazardous substances*

$$
\begin{bmatrix}
(x;1,1,1) & 1/VL & 1/VL & VH & L \\
VL & (x;1,1,1) & VL & VH & L \\
VL & 1/VL & (x;1,1,1) & H & M \\
1/VH & 1/VH & 1/H & (x;1,1,1) & H \\
1/L & 1/L & 1/M & 1/H & (x;1,1,1)
\end{bmatrix}.
$$



$$\tilde{p}_{14}^2 = (0.13, 0.25, 0.38), \quad \tilde{p}_{24}^2 = (0.17, 0.25, 0.50), \quad \tilde{p}_{34}^2 = (0.14, 0.16, 0.50), \quad \tilde{p}_{44}^2 = (0.07, 0.11, 0.18) \text{ and } \tilde{p}_{54}^2 = (0.05, 0.08, 0.16)$$

### *Seaport safety*

a) *Vessel safety*

$$\begin{bmatrix} (x;1,1,1) & (x;1,1,1) & 1/L & 1/L & 1/VH \\ (x;1,1,1) & (x;1,1,1) & VL & L & 1/H \\ L & 1/VL & (x;1,1,1) & 1/M & 1/H \\ L & 1/L & M & (x;1,1,1) & 1/M \\ VH & H & H & M & (x;1,1,1) \end{bmatrix}$$

$$\tilde{p}_{11}^3 = (0.06, 0.09, 0.16), \quad \tilde{p}_{21}^3 = (0.09, 0.14, 0.24), \quad \tilde{p}_{31}^3 = (0.06, 0.11, 0.18), \quad \tilde{p}_{41}^3 = (0.09, 0.16, 0.30) \text{ and } \tilde{p}_{51}^3 = (0.31, 0.49, 0.74).$$

b) *Traffic volume*

$$\begin{bmatrix} (x;1,1,1) & (x;1,1,1) & 1/L & 1/VL & 1/VH \\ (x;1,1,1) & (x;1,1,1) & L & M & 1/H \\ L & 1/L & (x;1,1,1) & 1/M & 1/M \\ VL & 1/M & M & (x;1,1,1) & 1/L \\ VH & H & M & L & (x;1,1,1) \end{bmatrix}$$

$$\tilde{p}_{12}^3 = (0.07, 0.12, 0.20), \quad \tilde{p}_{22}^3 = (0.11, 0.19, 0.31), \quad \tilde{p}_{32}^3 = (0.08, 0.14, 0.25), \quad \tilde{p}_{42}^3 = (0.09, 0.15, 0.31) \text{ and } \tilde{p}_{52}^3 = (0.22, 0.41, 0.69).$$

c) *Weather sea condition and channel condition*

$$\begin{bmatrix} (x;1,1,1) & (x;1,1,1) & 1/VL & (x;1,1,1) & (x;1,1,1) \\ (x;1,1,1) & (x;1,1,1) & L & M & L \\ VL & 1/L & (x;1,1,1) & L & (x;1,1,1) \\ (x;1,1,1) & 1/M & 1/L & (x;1,1,1) & 1/L \\ (x;1,1,1) & 1/L & (x;1,1,1) & L & (x;1,1,1) \end{bmatrix}$$

$$\tilde{p}_{13}^3 = (0.13, 0.18, 0.23), \quad \tilde{p}_{23}^3 = (0.17, 0.30, 0.48), \quad \tilde{p}_{33}^3 = (0.13, 0.21, 0.33), \quad \tilde{p}_{43}^3 = (0.07, 0.11, 0.20) \text{ and } \tilde{p}_{53}^3 = (0.14, 0.20, 0.31).$$

d) *Other safety factors*



$$\begin{bmatrix} (x;1,1,1) & VL & (x;1,1,1) & VL & VL \\ 1/VL & (x;1,1,1) & M & H & M \\ (x;1,1,1) & 1/M & (x;1,1,1) & M & L \\ 1/VL & 1/H & 1/M & (x;1,1,1) & 1/VL \\ 1/VL & 1/M & 1/L & VL & (x;1,1,1) \end{bmatrix}.$$

$\tilde{p}_{14}^{3} = (0.14, 0.18, 0.36)$, $\tilde{p}_{24}^{3} = (0.20, 0.37, 0.56)$, $\tilde{p}_{34}^{3} = (0.12, 0.21, 0.35)$, $\tilde{p}_{44}^{3} = (0.06, 0.11, 0.16)$ and $\tilde{p}_{54}^{3} = (0.08, 0.13, 0.24)$.

Preference indices of business processes under each identified criterion are calculated by using procedure (Step 3 of the proposed Algorithm). By using the proposed procedure (Step 5 to Step 7) the rank of business processes under evaluation criteria is determined.

The calculated preference indices of the treated business processes and their rank under the identified evaluation criteria are presented in the following text (Table 1, Table 2, Table 3).

Table 1 Preference indices of business processes and their rank under quality performance

Table 2 Preference indices of business processes and their rank under environmental protection performance

Table 3 Preference indices of business processes and their rank under safety performance

The overall preference index of each business process is calculated by using procedure (Step 4 of the proposed Algorithm). The rank of business processes with respect to all identified evaluation criteria and their weights and the degree of belief that a business process can be placed at first place in the rank are calculated (Step 4 to Step 7) and presented in Table 4.

Table 4 The overall preference index

**5 Discussion**

According to the final score, the business process (p=5) is the most preferred because it has the highest priority weight. The level of customers' satisfaction mostly depends on quality of this business process realization, so the obtained result is expected. With respect to the calculated degrees of belief, it can be suggested that the management team has defined an adequate reference model of an organization. At last place in the rank is business process (p=4). In the treated seaport, occupational health and environmental protection based on OHSAS 18001 standard, has been introduced recently. Some





activities related to occupational health and environmental protection are delegated to employees that have not been part of the management team. From this fact can be concluded that management team has not given a full commitment to new demands and it does not have enough knowledge so the assessment is obtained through previous experience.

In the course of determining the appropriate actions for performance enhancement within each identified business process, it

is necessary to present the sensitivity of each business process with respect to the KPIs and the main performances (Figs. 2 and 3). The values of performances and KPIs are presented by crisp values. These values are obtained by using a defuzzification method (Zimmermann, 2001).

Figure 2 – Sensitivity of each business process with respect to the KPIs

Figure 3 – Sensitivity of each business process with respect to the performances

Business process *(p=1)* is the most sensitive with respect to quality performance. Since customers represent end users of seaport services, a low level of quality of the treated business process will decrease profit. KPIs that generate the highest impact within this performance are *Average number of customers* and *Average number* of vessels in the queue. Management initiatives which could lead to the enhancement of the denoted KPIs are application quality methods (QFD, cost-benefit analysis, Define Measure Analyze Control (DMAC), etc.).

Business process *(p=2)* is the most sensitive with respect to environmental protection. In relation to the conducted activities during this process's realization (maintenance of vessels, port transportation, cleaning, garbage and hazardous substance disposal, etc.), the quality of air and the quality of water could be decreased and generation of noise and leaking of hazardous substances could be increased. It may be concluded that all KPIs are almost equally important. Management initiatives that should lead to KPI values' enhancement should cover activities of the definition of procedures that are based on international

standards and directives. Other activities could be oriented to the training of employees.

Business process *(p=3)* is the most sensitive in terms of environmental protection performance. Endangering the environment occurs during the implementation of maintenance dock activities and cranes, as well as other transport manipulating systems, warehouses, roads, etc. The most significant KPIs in the scope of this analysis are *Water quality* and *Noise*. The management initiatives that should lead to KPI values' enhancement correspond to process *(p=2)*.

In terms of sensitivity, business process *(p=4)* is mostly impacted by safety performance. A low level of safety performance may cause a decreased level of a seaport's competitiveness. KPIs that generate the highest impact within this performance are vessel safety and traffic volume. The management initiatives that should lead to the treated KPI values' enhancement could be the definition of procedures for safety optimization of vessel pilotage and optimization of the volume of traffic inside a port. These procedures should be in compliance with international standards and directives.

When business process *(p=5)* is analysed, safety performance makes the most significant impact in terms of sensitivity. Most activities generated by this process are customer oriented so low performances of this process could lead to a decrease of competitiveness and a bad image of the port. Enhancement of this process with respect to safety performance may be achieved by applying the measures for enhancement of vessel safety and traffic volume which are already explained.



## 5.1 Research Implications

By comparing papers which propose a model for evaluating business processses under uncertainties, certain differences could be noted, which are further described. This analysis, at the same time, shows the advantages of the proposed model.

In this paper, it is assumed that determination of the relative importance of the performance of business processes and the relative preference of KPIs of performances is more reliable when obtained using pair-wise comparison than when they are directly obtained, because it is easier to make a comparison between two criteria than make an overall weight assignment. The fuzzy pair-wise comparison matrices of the relative importance of performances and their KPIs are constructed. The elements of these matrices are given by using the FOWA aggregation method. The weights vector of performances, weights vector of KPIs under each performance and preferences vector of business processes under each KPI are calculated by using the method which is developed in (Wu et al., 2004). In this way, the fuzzy ranks of business processes are given. It can be denoted as the main difference between this paper and the papers which can be found in the literature (Tadic et al., 2013; Pak et al., 2015; Kaya and Kahraman, 2011; Hsu, 2012).

The overall index of the preferences for each business process is described by a fuzzy number. According to fuzzy algebra rules, values of the overall index of a preference are not TFNs but it is possible to express approximated values of fuzzy operations as TFNs (Kwong and Bai, 2003). Therefore, according to the overall index of a preference, the ranking order of all business processes can be determined and the most important one from among a set of treated business processes can be selected. The degree of belief that any business process can be the business process which is associated with the highest value of the overall index of a preference can be determined. The priority of management initiatives that should lead to enhancement of business processes should be based on the rank of business processes and the calculated degree of beliefs. The introduced modifications in determining priority of management measures represents the main difference, and at the same time, the advantage of the proposed model compared to the proposed FAHP methods which can be seen in the literature.

## 6 Conclusion and future work

Seaport management practice shows that evaluation and enhancement of business processes represent one of the most relevant issues of competitiveness and sustainability. Definition of an enhancement strategy should be based on the rank of the business processes. The main performances and their KPIs are determined in compliance with the process approach and ISO 20000-1:2000. A large number of decision variables demonstrating the complexities are involved in the ranking of business processes. It is assumed that application of analytic methods in determining the rank of business processes is better than applying intuitive decision making methods. It may be suggested that each solution obtained in an exact way is less encumbered by the subjective views of decision makers so this could make it more accurate.

A modified fuzzy extended AHP (MFAHP) is proposed. Uncertainties in the relative importance of each pair of performances and their KPIs and the preference of business processes with respect to each identified KPI are described by



pre-defined linguistic expressions which are modelled by using fuzzy sets theory. The fuzzy approach is easy to understand and flexible and it is tolerant to imprecise data. These linguistic expressions are modelled by TFNs.

Evaluation of the relative importance of business process performances and their KPIs is based on knowledge, experience of the seaport management team, needs of local government and other stakeholders. Applying fuzzy group decision making in

determining these decision variables can be considered as one of the contributions of this paper. The main contribution of this paper may be seen as an application of the proposed MFAHP with a goal to obtain the fuzzy rank of business processes and the degree of belief that a business process can be placed at first place. With respect to the fuzzy rank and degrees of belief, it is possible to rationalize expenditure of time, money and other resources. Also, a good scheduling of management initiatives' order could increase efficiency of the enhancement strategy. This can be considered as the main contribution of

the proposed MFAHP which was tested with real life data and the obtained results are presented.

The main advantages of the proposed MFAHP are related to the fact that it does not involve cumbersome mathematical operations and could be easily employed within seaports which operate in an uncertain environment. The proposed MFAHP can be easily extended to the analysis of other management decision problems in different research areas. The general limitations of the model are the need for well structured business processes and comprehensible definition of their

performances.

Finally, it is clear that further research could cover a more detailed decomposition of business processes, an increased number of performances and their KPIs, and connection of the business processes of the treated seaport with business processes of other seaports in similar regions.

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





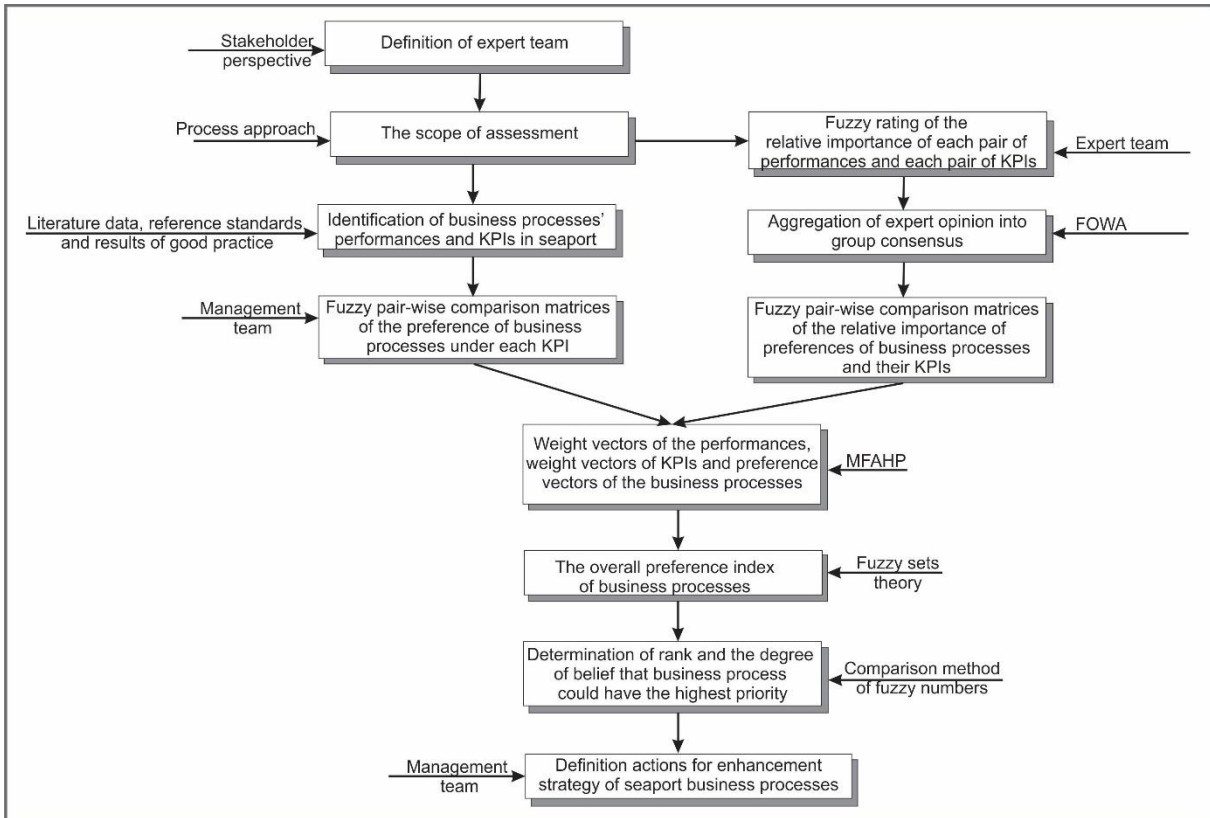

**Figure 1 – The evaluation procedure of seaport business processes by MFAHP**





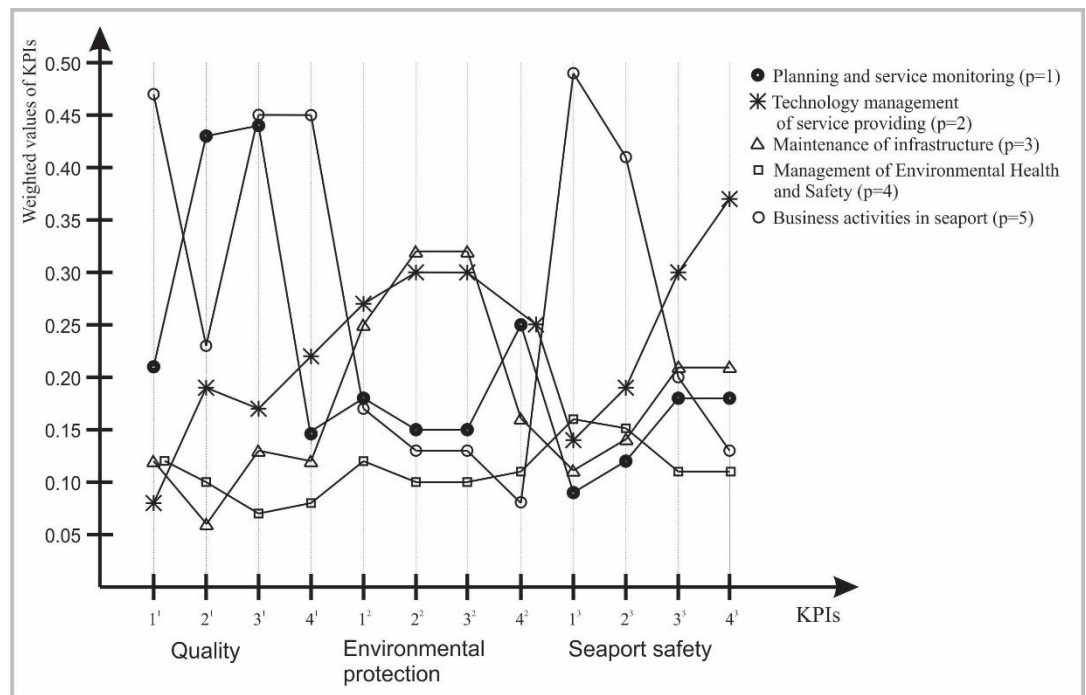

**Figure 2 – Sensitivity of each business process with respect to the KPIs**

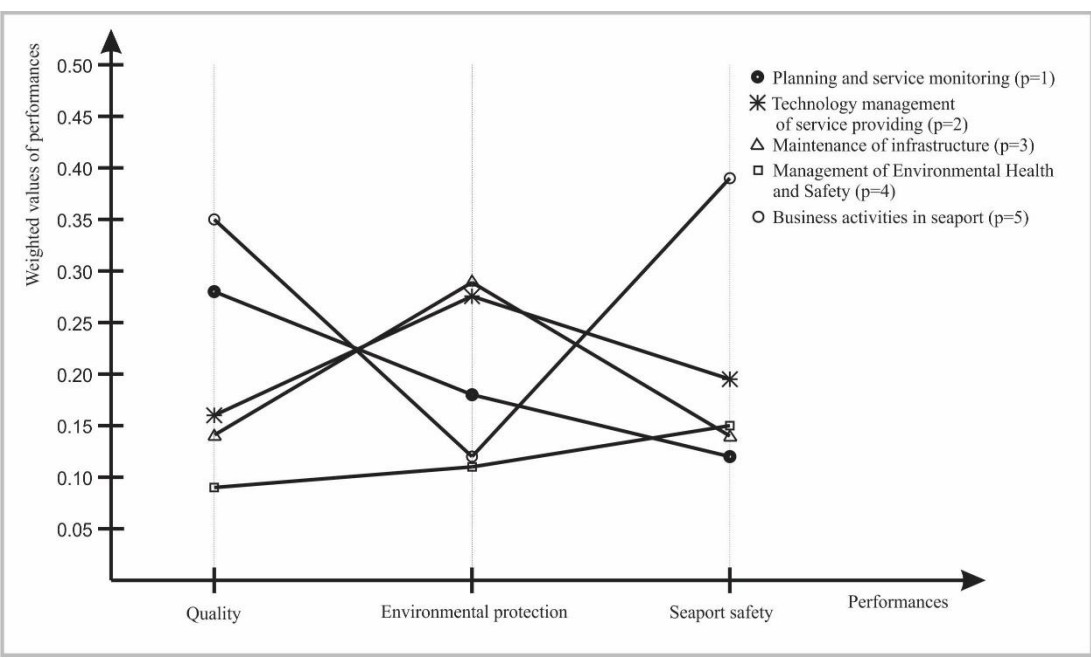

5    **Figure 3 – Sensitivity of each business process with respect to the performances**



Table 1 Preference indices of business processes and their rank under quality performance

| Process no. | Preference index | Rank | Degree of belief that business process can be the best |
|---|---|---|---|
| p=1 | (y; 0.120, 0.284, 0.658) | 2 | 0.889 |
| p=2 | (y; 0.075, 0.163, 0.423) | 3 | 0.599 |
| p=3 | (y; 0.043, 0.141, 0.257) | 4 | 0.347 |
| p=4 | (y; 0.044, 0.095, 0.237) | 5 | 0.262 |
| p=5 | (y; 0.147, 0.348, 0.755) | 1 | 1 |

Table 2 Preference indices of business processes and their rank under environmental protection performance

| Process no. | Preference index | Rank | Degree of belief that business process can be the best |
|---|---|---|---|
| p=1 | (y; 0.065, 0.181, 0.490) | 3 | 0.786 |
| p=2 | (y; 0.109, 0.282, 0.872) | 2 | 0.991 |
| p=3 | (y; 0.094, 0.289, 0.831) | 1 | 1 |
| p=4 | (y; 0.038, 0.106, 0.305) | 5 | 0.536 |
| p=5 | (y; 0.040, 0.122, 0.329) | 4 | 0.585 |

Table 3 Preference indices of business processes and their rank under safety performance

| Process no. | Preference index | Rank | Degree of belief that business process can be the best |
|---|---|---|---|
| p=1 | (y; 0.056, 0.121, 0.289) | 5 | 0.319 |
| p=2 | (y; 0.083, 0.199, 0.472) | 2 | 0.617 |
| p=3 | (y; 0.058, 0.143, 0.349) | 4 | 0.540 |
| p=4 | (y; 0.059, 0.145, 0.397) | 3 | 0.489 |
| p=5 | (y; 0.162, 0.391, 0.879) | 1 | 1 |



Table 4 The overall preference index

| Process no. | The overall preference index | Rank | Degree of belief that business process can be the best |
|---|---|---|---|
| p=1 | (y; 0.054, 0.195, 0.695) | 3 | 0.819 |
| p=2 | (y; 0.056, 0.199, 0.775) | 2 | 0.841 |
| p=3 | (y; 0.039, 0.167, 0.588) | 4 | 0.755 |
| p=4 | (y; 0.033, 0.119, 0.476) | 5 | 0.649 |
| p=5 | (y; 0.089, 0.329, 1.101) | 1 | 1 |

