# Peer review of "The evaluation and enhancement of quality, environmental protection and safety in seaports"

_Natural Hazards and Earth System Sciences, 2016_

## Referee Comment (RC1) · Anonymous Referee #1 · 19 May 2016

**General comments**

The manuscript presents an application of Analytic Hierarchy Process (AHP, (Saaty, 2008)) for the ranking of "business processes" in the context of harbour management. Qualitative expert judgements are represented in terms of triangular fuzzy numbers (TFN), i.e. triples of conventional (or: crisp) numbers. Derived weights of the pairwise comparison judgements are employed for ranking priorities for the harbour management processes.

AHP has been successfully applied to various cases of multicriteria optimization problems, such as strategic and military actions, customer satisfaction, development of new products (Saaty, 2008). In this respect, I find interesting the attempt by the authors to use AHP in combination with fuzzy numbers for modeling the decisional process in a harbour. However, in order to accomplish the goal of an enhanced objectiveness and

transparency, AHP should be employed using high methodological standards and with a clear analysis of the primary sources of judgment. This is actually the main criticism I feel to move to the paper, that in fact ends up with the opposite result of making the decisional process in the harbour even more obscure and more subjective than without AHP. My concern is declined into two major issues that I have with the manuscript. Formally, it is not well organised, making it quite hard to read and verify. Concerning the contents, I find the methodology not completely sound and not capable of supporting the main conclusions. I will address both issues in detail in the specific and technical comments below.

I think that in order the manuscript to match the minimum requirements for publication in a peer reviewed journal, the authors should first analytically address all issues listed in the following.

**Specific comments**

**A - Formal structure**

A1  Acronyms.
Several acronyms are used without any previous definition, such as: QMS, TQM, FAHP (given 13 lines after its first use in terms of AHP), AHP (that, apart from a reference within the abstract to "modified fuzzy extended analytic hierarchy process" or MFAHP, is nowhere directly defined), TFN (given just in the abstract, where it is not used, but not in rest of the paper), APQC.

A2  Organization.
The review of literature is spread among different sections and this is not justified by the reference to it done in the rest of the contents. Literature reports appear

not just in the introduction (Sect.1) but also in the initial part of Sect.2 called "Materials and methods" and in whole Sect.3. The material in Sect.3 could be introduced before Sect.2.1. The main algorithm is kind of repeated in two versions: one on P4-5 and the other on P6-7. Sect.3 contains a description of KPIs that would deserve an indentation. Furthermore, no reference to the application done later on, matrices at P12-15, is done, where instead a)-e) letters are employed for sorting the various KPIs. I suggest to use the same letters in the list of KPI in Sect.3.

A3 Notation.

-Sect.2.2 is highly repetitive and does not help in reading and memorising key quantities. I suggest to replace the contents of Sect.2.2 with a table as Tab. 1 of this review and to simplify the symbol for the fuzzy numbers: do not use $x$ or $y$ and just give the triple of crisp numbers making the TFN. E.g. $(x; 2, 3, 4) \rightarrow (2, 3, 4)$.

- Most of these symbols introduced on Sect.2.1 (e.g. $\varepsilon, \kappa, \varphi_k, \iota, E, K, J_k, I$) are not at all or just poorly used in the following of the manuscript.

- there is a confusing nomenclature about "weights vector of performance", "weights vector of KPI", and "preference vector of business process".

- Since I do not see any reason for breaking the alphabetical order, I would replace $\chi$ with $\gamma$ in Eq.2,3 at P6

**Table 1.** Suggested table to replace material in Sect.2.2. Please consider note on "business processes" expression in A4 item.

|  | set symbol | running index | set size symbol | set size |
|---|---|---|---|---|
| experts | $\varepsilon$ | $e$ | $E$ | 4? |
| performances | $\kappa$ | $k$ | $K$ | 3 |
| KPI of $k$th performance | $\varphi_k$ | $j$ | $J_k$ | 4 |
| "business processes" | $\iota$ | $i$ | $I$ | 5 |

A4 Other.

- do not use the word "business" both in the collective expression "business process" and for one of its actual implementations ($p = 5$: "business activities in seaport")! This is a highly confusing linguistic choice made by the authors, I really cannot approve it.

- write matrices at P11-15 as equations whose l.h.s. is some meaningful combination of symbols with pedices or apices related to the actual contents of the matrix (consider symbols introduced in Tab. 1 of this review)

- Fig.1 is quite complex and not entirely related to the text. It could be simplified, highlighting (i.e., numbering) the steps of the proposed methodology;

- caption of Fig.2 could explain more directly that the horizontal axis contains the performances, detailed per KPI. Also, the notation $1^1, 2^1, ...$ is quite confusing at first sight.

- Sentence at the end of P4 ("Value 1, and value 0 denote that one performance or KPI is as important, or unimportant, as any identified performances or KPIs under each treated performance") does not add any understanding and can be removed.

A5 Figures and Tables.

The list of processes in the legend of both Fig.1 and 2 is referenced both in Tab.1,2,3 and in the manuscript. Thus, it deserves an independent presentation in a specific table.

A6 English.

Specific sentences are really badly formulated. E.g. "In the course of easier understanding of the proposed Algorithm, in this Section the notation is given" (P5, row14). Revision by a professional translator of technical manuscripts is

highly recommended.

**B - Actual Contents**

B1 Abstract.
The proposed model is far from being "verified", demonstrated or validated within this paper. Instead a simple numerical evaluation of the "proposed algorithm" is carried out. Furthermore, the conclusions are quite surprising, see item B5.

B2 Problem statement and methodology.
- First of all, see all comments done in **A**, since the actual scientific contents of a paper can be hardly detached from their presentation style.
- It should be more clearly stated what the input data for all subsequent elaborations are. In particular, the weights $w_e = (.4, .3, .2, .1)$ of the experts used are present in the example of line 14 of P10. I -and I think most readers too- would like to see a table where these weights are clearly associated to the 4 experts (not sure if in this order, but they seem to be: seaport owner, main manager, local government, operational management of the seaport).
-Furthermore, the most influential expert overweights by 4 times the least influential one. How were the $w_e$ assessed? This raises the more fundamental question "who is judging the judges?". This information about expert judgement is quite crucial for the actual numerical outcomes, see B5.
- as from the definitions of the base TFNs (P4), the authors use a linear scale $[\frac{1}{\sigma}, \sigma]$ with $\sigma = 5$. The type of scale (Ishizaka and Labib, 2009) and the quantity $\sigma$ are keys in a pairwise comparison matrix, representing the accuracy of the judgements and indirectly affecting matrix consistency, see e.g. (Ramík, 2009). It is usually taken $\sigma = 9$ (Saaty, 2008). In my opinion, the actual choice of the quality and extent of the scale deserves a dedicated comment by the authors.

[Figure]

- why are there so many crisp numbers $(1, 1, 1)$ in the off-diagonal elements of the pairwise comparison matrixes at P11-15 ? The authors make a big point about modeling uncertainty in terms of fuzzy numbers, and then it turns out that several specific processes can be assessed to have exactly the same relative importance (such is in fact the meaning of $(1, 1, 1)$ in the matrixes). I find it odd that there is not even a comment on this.

B3 Pairwise comparison matrices.
- The numerical case study (Sect. 4) starts all of a sudden with a pairwise comparison matrix, whose relevance to the method (which is great) is never mentioned but in Fig.1.
- The consistency of this matrix (Ramík, 2009) is never evaluated nor discussed. Given the qualitative nature of the expert judgements, consistency is a quite relevant concern of an AHP investigation (Saaty, 2008). Thus, I believe some measure of consistency should be computed and provided for all comparison matrixes in the manuscript. E.g. is the consistency ratio below the classical threshold of 10%?

B4 Missing originality.
- The specificity of the claimed "modified" FAHP (MFAHP) method proposed by the authors is not demonstrated nor stated. The core of the proposed algorithm (steps # 5-8 of Sect.2.1) is just a few standard rules taken from the literature, while the rest (steps # 1-4 of Sect.2.1) is just definitions. Unless the authors clearly state where the originality of the proposed algorithm is, I think they cannot claim to have developed a new method: they just made an application of an existing one, and the use of the dedicated acronym MFAHP is not justified, in my opinion.

B5 Not fully justified conclusions.
- According to Tab.1 and Tab.3 the "business activities in seaport" process ($p = 5$)

gets rank 1 for both the quality and the safety performance. How can a business activity be the most crucial action for enhancing safety of a harbour? The authors comment this surprising finding by stating that "the level of customers' satisfaction mostly depends on quality of this business process realisation, so the obtained result is expected" (P15, rows 23-24). I actually thought that the focus of the paper was to establish priorities for the port management without a specific perspective on customers, but in view of multi-criteria optimization. If instead the authors mean that the whole analysis is just functional to enhance customers' satisfaction, then the title, abstract and scope of the paper should be consequently restricted. In any case, I cannot easily accept that business activities will enhance safety of a harbour. I think that either there is some numerical manipulation mistake or the initial expert assessments (including their relative weights) were biased. This leads me back to the observation about expert weights (B2) and missing analysis of consistency of the pairwise comparison matrices (B3).

**Technical comments**

C1 For a symmetry reason, on P4 it seems to me much more natural to define VL=(1,2,2) and not VL=(1,1,2): just plot the 5 fuzzy numbers VL, L, M, H, VH and see why. Actually it would help the reader in having this plot as a Figure of the manuscript.

C2 Matrix on P10, row 10 (please, use symbols for identifying mathematical objects more easily!):
- I guess the $3 \times 3$ matrix refers to the $K = 3$ performances and each fuzzy number

in the 4-tuples refers to an expert judgement. If this is correct, it should be clearly stated. Furthermore, for consistency of notation, the diagonal elements should be 4-tuples of crisp numbers, something like $(1,1,1),(1,1,1),(1,1,1),(1,1,1)$ that could be conveniently replaced by a convenient multi-dimensional identity symbol such as the one expressed in LaTeX by \mathbb{1}.
- In the following, (P12-15), also $4 \times 4$ (P11) and $5 \times 5$ appear. It would be good to always state what this dimensionality refers to. I suppose that they refer to $J_k = 4$ KPIs of each performance, and to the $I = 5$ "business processes", see Tab. 1 of this review.

C3 - on P4, row21: replace "consensus" by "group consensus" and make reference to Step 5 (P5) of the algorithm.
- it is unnecessary to define again $\tilde{W}^e_{kk'} = (...)$ and $\tilde{W}^e_{jj'} = (...)$ on P6, row10, after they were introduced in Sect2.2
- remove range of indexes ($i = ...j = ...k = ..$) in both Eq.(4) and Eq.(5): they were already introduced in Sect.2.2;
- Eq.(5) could be better rewritten as

$$\tilde{a}_i = \sum_{k=1}^{K} \tilde{w}_k \, \tilde{a}^k_i = \sum_{k=1}^{K} \sum_{j=1}^{J_k} p^k_{ij}$$

- there is a logical need to insert a separation (new subsection) on P10, row8.

**References**

Ishizaka, A. and Labib, A.: Analytic hierarchy process and expert choice: Benefits and limitations, OR Insight, 22, 201–220, 2009.

[Figure]

Merigó, J. M. and Casanovas, M.: Using fuzzy numbers in heavy aggregation operators, International Journal of Information Technology, 4, 177–182, 2008.

Ramík, J.: Consistency of Pair-wise Comparison Matrix with Fuzzy Elements., in: IFSA/EUSFLAT Conf., pp. 98–101, 2009.

Saaty, T. L.: Decision making with the analytic hierarchy process, International journal of services sciences, 1, 83–98, 2008.

---

## Author Comment (AC1) · 20 Jun 2016

Responses to reviewer's comments on "The evaluation and enhancement of quality, environmental protection and safety in seaports", by D. Tadic et al., are given in a pdf document of the zipped dir.zip in supplement. All changes are incorporated into revised manuscript (denoted in red) and revised manuscript is also given as pdf file in supplement.

Please also note the supplement to this comment:
http://www.nat-hazards-earth-syst-sci-discuss.net/nhess-2016-126/nhess-2016-126-AC1-supplement.zip

2016.

---

## Editor Comment (EC1) · PhD Archetti (Editor) · 8 Jul 2016

Please can you post the Responses to reviewer's comments on the open discussion.

---

## Author Comment (AC2) · 27 Jul 2016

article [utf8]inputenc

array color amsmath

**Response to the reviewer's comment**

Authors would like to thank the reviewer for constructive comments (italic), relevant suggestions and corrections. All changes are incorporated into revised manuscript and denoted in red.The revised manuscript is given as a pdf document in supplement. We remain at your disposal for any further information.

**Anonymous reviewer - General comments**

*The manuscript presents an application of Analytic Hierarchy Process (AHP, (Saaty, 2008)) for the ranking of "business processes" in the context of harbour management. Qualitative expert judgements are represented in terms of triangular fuzzy numbers (TFN), i.e. triples of conventional (or: crisp) numbers. Derived weights of the pairwise comparison judgements are employed for ranking priorities for the harbour manage-ment processes.*
*AHP has been successfully applied to various cases of multicriteria optimization prob-lems, such as strategic and military actions, customer satisfaction, development of new products (Saaty, 2008). In this respect, I find interesting the attempt by the authors to use AHP in combination with fuzzy numbers for modeling the decisional process in a harbour. However, in order to accomplish the goal of an enhanced objectiveness and transparency, AHP should be employed using high methodological standards and with a clear analysis of the primary sources of judgment. This is actually the main criticism*

*I feel to move to the paper, that in fact ends up with the opposite result of making the decisional process in the harbour even more obscure and more subjective than without AHP. My concern is declined into two major issues that I have with the manuscript. Formally, it is not well organised, making it quite hard to read and verify. Concerning the contents, I find the methodology not completely sound and not capable of supporting the main conclusions. I will address both issues in detail in the specific and technical comments below.*

*I think that in order the manuscript to match the minimum requirements for publication in a peer reviewed journal, the authors should first analytically address all issues listed in the following.*

**Anonymous reviewer: A- Formal structure**

*A1 - Acronyms.*

*Several acronyms are used without any previous definition, such as: QMS, TQM, FAHP (given 13 lines after its first use in terms of AHP), AHP (that, apart from a reference within the abstract to "modified fuzzy extended analytic hierarchy process" or MFAHP, is nowhere directly defined), TFN (given just in the abstract, where it is not used, but not in rest of the paper), APQC.*

**Response:**

- A Quality Management System (QMS) conforming to ISO 9001:2008 should be considered as an important additional step, in terms of quality, because ISO 9001 also takes into account economic and financial aspects, design and development aspects, and introduces a management review for measurement and analysis

of a process with the aim of improving performances (Poli et al., 2012). However, this important issue forces every organization to start either with ISO 9000 or Total Quality Management (TQM) as a business strategy (Sedani and Lakhe, 2011).

The number and type of business processes in a seaport is defined with respect to American Productivity and Quality Center (APQC) Process Classification Framework (PCF) and process owner opinion.

The mentioned integration includes: a) presentation of a seaport as a network of unrelated business processes so the overall success of the business processes may be assessed on the level of predefined criteria; b) the assessment of business processes by fuzzy Analytic Hierarchy Process (FAHP); c) definition of management initiatives which should lead to the improvement of business success; the order of taking management initiatives is based on the obtained rank of business processes.

Experts and operational managers use the pre-defined linguistic expressions, which are modelled by triangular fuzzy numbers (TFNs).

**A2 - Organization.**

*The review of literature is spread among different sections and this is not justified by the reference to it done in the rest of the contents. Literature reports appear not just in the introduction (Sect.1) but also in the initial part of Sect.2 called "Materials and methods" and in whole Sect.3. The material in Sect.3 could be introduced before Sect.2.1. The main algorithm is kind of repeated in two versions: one on P4-5 and the other on P6-7. Sect.3 contains a description of KPIs that would deserve an indentation. Furthermore, no reference to the application done later on, matrices at P12-15, is done, where instead a)-e) letters are employed for sorting the various KPIs. I suggest to use the same letters in the list of KPI in Sect.3.*

**Response:**

- References:
  David, F.: Strategic Management, Upper Saddle River, N.J. USA: Prentice Hall-Pearson, 2011.
  Hutchins, D.: Hoshin Kanri: The Strategic Approach to Continous Improvement. England: Gower e-Book, 2008.
  have been removed from the manuscript.
  Text:
  "The seaport operations may be described with a lot of uncertainties, so lately there have been many papers in literature that deal with risk management models (John et al., 2014) and metrics, proposed and numerically implemented to assess the overall performance of large systems, during natural disasters and their recovery – resilience (Shafieezadeh and Burden, 2014). This is due to the fact that much of the available data associated with port operations require a flexible but robust approach of handling as well as updating existing information with new data. As risk management activities are oriented to safety, port safety evaluation (Pak et al., 2015) is the first step in overall safety enhancement. After quality management certification, determining of performances of business processes is based on pre-defined critical success factors (CSFs) (Oakland, 2004)."

  has been moved to the section 1 in the revised manuscript.

  Section 2 of the revised manuscript has been renamed to 2 Analysis of performances, key performances indicators and business processes in a seaport.

  Section 2.1 has been renamed to 3. The model for evaluation of seaport business processes in the revised manuscript. Also, this section has been improved in a manner that the proposed algorhytm is not repeated as it has been

suggested.

As the reviewer suggested, in order to make the reference with the application, the identified key performance indicators have been denoted as it is presented.

(Q1) Quality of the seaport services
(Q2) Average number of customers
(Q3) Average number of vessels in the queue
(Q4) Pilotage operation of the vessel
(E1) Quality of air
(E2) Water quality and (E3) Noise
(E4) Hazardous substances
(S1) Vessel safety
(S2) Traffic volume
(S3) Weather sea condition and channel condition
(S4) Other safety factors

**A3 - Notation.**

- *Sect.2.2 is highly repetitive and does not help in reading and memorising key quantities. I suggest to replace the contents of Sect.2.2 with a table as Tab. 1 of this review and to simplify the symbol for the fuzzy numbers: do not use x or y and just give the triple of crisp numbers making the TFN. E.g. $(x; 2, 3, 4) \rightarrow (2, 3, 4)$ .*

- *Most of these symbols introduced on Sect.2.1 (e.g. $\varepsilon, \kappa, \varphi_k, \iota, E, K, J_k, I$) are not at all or just poorly used in the following of the manuscript.*

- *there is a confusing nomenclature about "weights vector of performance", "weights vector of KPI", and "preference vector of business process".*

- *Since I do not see any reason for breaking the alphabetical order, I would replace $\chi$ with $\gamma$ in Eq.2,3 at P6.*

**Table 1** *Suggested table to replace material in Sect.2.2. Please consider note on "business processes" expression in A4 item.*

|  | set symbol | running index | set size symbol | set size |
|---|---|---|---|---|
| experts | $\varepsilon$ | $e$ | E | 4? |
| performances | $\kappa$ | $k$ | K | 3 |
| KPI of the $k$th performance | $\varphi_k$ | $j$ | $J_k$ | 4 |
| "business processes" | $\iota$ | $i$ | I | 5 |

**Response:**

- The notation has been formatted into table.
  Table 2 notation

|  | running index | set size symbol | set size |
|---|---|---|---|
| heightexperts | e | E | 4 |
| heightperformances | k | K | 3 |
| heightKPI of the kth performance | j | $J_k$ | 4 |
| heightbusiness process height | i | I | 5 |

- The structure (x; 2; 3; 4) has been transformed into (2; 3; 4).

  *very low importance/preferency*: VL=(1,1,2)
  *low importance/preferency*: L=(1,2,3)

*moderate importance/preferency*: M=(2,3,4)
*high importance/preferency*: H=(3,4,5)
*very high importance/preferency*: VH=(4,5,5)

- All unnecessary symbols have been removed in the revised manuscript.

- The alphabetical order is respected in revised manuscript.

**A4 - Other.**

- *do not use the word "business" both in the collective expression "business process" and for one of its actual implementations (p = 5: "business activities in seaport")! This is a highly confusing linguistic choice made by the authors, I really cannot approve it.*

- *write matrices at P11-15 as equations whose l.h.s. is some meaningful combination of symbols with pedices or apices related to the actual contents of the matrix (consider symbols introduced in Tab. 1 of this review)*

- *Fig.1 is quite complex and not entirely related to the text. It could be simplified, highlighting (i.e., numbering) the steps of the proposed methodology;*

- *caption of Fig.2 could explain more directly that the horizontal axis contains the performances, detailed per KPI. Also, the notation $^1, 2^1, ...$ is quite confusing at first sight.*

- *Sentence at the end of P4 ("Value 1, and value 0 denote that one performance or KPI is as important, or unimportant, as any identified performances or KPIs under each treated performance") does not add any understanding and can be removed.*

**Response:**

- Adjective business has been used with processes. Sub process Business activities in seaport (p=5) has been changed into Activities in seaport (p=5).

- Matrices P11-P14 have been modified in compliance with reviewer's suggestion.

- Figure 1 has been simplified in revised manuscript.

  Fig. 1. The evaluation procedure of seaport business processes by FAHP
  (Please see the Fig. 1 below)

- Figure 2 has been improved in the terms of notation for better understanding in the revised manuscript.

  Fig. 2. Sensitivity of each business process with respect to the KPIs

  (Please see the Fig. 2 below)

- The sentence has been removed.

***A5 - Figures and tables.***

*The list of processes in the legend of both Fig.1 and 2 is referenced both in Tab.1,2,3 and in the manuscript. Thus, it deserves an independent presentation in a specific table.*

**Response:**

The table of processes has been incorporated into revised manuscript.

Table 1 Identified processes in the seaport

| Running index | Title of the business process |
|---|---|
| heightp=1 | Planning and service monitoring |
| heightp=2 | Technology management of service providing |
| heightp=3 | Maintenance of infrastructure |
| heightp=4 | Management of Environmental Health and Safety |
| heightp=5 | Activities in seaport |

**A6 - English.**

*Specific sentences are really badly formulated. E.g. "In the course of easier under-standing of the proposed Algorithm, in this Section the notation is given" (P5, row14). Revision by a professional translator of technical manuscripts is highly recommended.*

**Response:**

English has been improved.

**Anonymous Reviewer: B - Actual Contents**

**B1 - Abstract.**

*The proposed model is far from being "verified", demonstrated or validated within this paper. Instead a simple numerical evaluation of the "proposed algorithm" is carried out. Furthermore, the conclusions are quite surprising, see item B5.*

**Response:**

- The term verified is replaced in the revised manuscript.

- model is tested through an illustrative example with real life data, where the obtained data suggest measures which should enhance business strategy and improve key performance indicators.

**B2 - Problem statement and methodology.**

- *First of all, see all comments done in A, since the actual scientific contents of a paper can be hardly detached from their presentation style.*

- *It should be more clearly stated what the input data for all subsequent elaborations are. In particular, the weights $w_e = (.4, .3, .2, .1)$ of the experts used are present in the example of line 14 of P10. I -and I think most readers too- would like to see a table where these weights are clearly associated to the 4 experts (not sure if in this order, but they seem to be: seaport owner, main manager, local government, operational management of the seaport).*

- *Furthermore, the most influential expert overweights by 4 times the least influential one. How were the $w_e$ assessed? This raises the more fundamental question "who is judging the judges?". This information about expert judgement is quite crucial for the actual numerical outcomes, see B5.*

- *as from the definitions of the base TFNs (P4), the authors use a linear scale $[1/\sigma, \sigma]$ with $\sigma = 5$. The type of scale (Ishizaka and Labib, 2009) and the quantity $\sigma$ are keys in a pairwise comparison matrix, representing the accuracy of the judgements and indirectly affecting matrix consistency, see e.g. (Ramík, 2009). It is usually taken $\sigma = 9$ (Saaty, 2008). In my opinion, the actual choice of the quality and extent of the scale deserves a dedicated comment by the authors.*

- *why are there so many crisp numbers (1,1,1) in the off-diagonal elements of the pairwise comparison matrixes at P11-15 ? The authors make a big point about modeling uncertainty in terms of fuzzy numbers, and then it turns out that several specific processes can be assessed to have exactly the same relative importance (such is in fact the meaning of (1,1,1) in the matrixes). I find it odd that there is not even a comment on this.*

**Response:**

- All comments defined in the part A have been incorporated into revised manuscript.

- Based on the internal policy of treated seaport, the expert team is adjoined with different specific weights (table 3).

  Table 3 - Specific weights of expert team

  | Experts | Specific weight of the expert |
  |---|---|
  | heightseaport owner | 0.4 |
  | heightmain manager | 0.3 |
  | heightlocal government expert | 0.2 |
  | heightThe representative of operational management of the seaport | 0.1 |

- We want to thank to the reviewer for this very useful comment. However, the proposed model is tested in one seaport in the process of restructuring in developing country. Our truthful intention was to describe the real situation so we had similar questions (like reviewer) but we have decided to stick with the real situation.

- According to Ishizaka and Labib, (2009), the verbal comparison must be converted into numerical scales, such as linear (Power, Geometric, Logarithmic, etc.). Also, mentioned authors have concluded that "Theoretically there is no reason to be restricted to these numbers and verbal gradation."In the revised manuscript, we have decided to proceed like Chang (1996). The domains of fuzzy numbers can be defined on different scales (Ishizaka and Labib, 2009) and in this paper the domains of presented TFNs are defined into interval [1-5].
  Chang, D., Y.: Applications of the extent analysis method on fuzzy AHP, European Journal of Operational Research, 95, 649-655, doi:10.1016/0377-2217(95)00300-2, 1996.

- All judgements were made by experts and authors came to the similar conclusion as a reviewer. There are some crisp numbers (1, 1, 1) in the off-diagonal elements but we wanted to present the real state and opinion of experts.

**B3 - Pairwise comparison matrices.**

- *The numerical case study (Sect. 4) starts all of a sudden with a pairwise comparison matrix, whose relevance to the method (which is great) is never mentioned but in Fig.1.*

- *The consistency of this matrix (Ramík, 2009) is never evaluated nor discussed. Given the qualitative nature of the expert judgements, consistency is a quite relevant concern of an AHP investigation (Saaty, 2008). Thus, I believe some measure of consistency should be computed and provided for all comparison matrixes in the manuscript. E.g. is the consistency ratio below the classical threshold of %?*

**Response:**

- The presence of the pairwise comparison matrix has been emphasized in the figure 1 in revised manuscript.

- Thank you for the very useful suggestion. We have calculated consistency of the matrices and expert team did the assessment again, more carefully. Improvement of the revised manuscript are following:
  Fuzzy pair-wise comparison matrices of the relative importance of performance, the relative importance of KPI under each performance and preference of business processes respecting each KPI are stated. Before all the calculation of vectors of priorities it is necessary to determine the coefficient of consistency to reflect the consistency of the decision makers' judgements during the evaluation phase (Saaty, 2008). Calculation of consistency may be delivered by using the method of logarithmic least squares (Lootsma, 1996), eigen vector method (Saaty, 2008), method of geometric mean (Ramik, 2009), etc. The eigen vector method represents a natural measure for inconsistency and it is used in wide literature and it is used in this paper, too. It is worth to mention that all relevant indexes of consistence (C.I.) should be equal or below the threshold of 0.1.
  The elements of constructed fuzzy pair-wise matrices are defuzzified, and after that, the consistence of fuzzy pair-wise matrices is determined. It is determined by analogy with Torfi et al., (2010).

Please note the pdf document given as an addition to the B3 response and attached as a separate pdf file in the compressed supplement to this comment, where the fuzzy pair-wise comparison matrices of the relative importance of performance, the relative importance of KPI under each performance and preference of business processes respecting each KPI are stated.

- Preference indices of business processes under each identified criterion are cal-
  culated by using procedure (Step 3 of the proposed Algorithm). By using the
  proposed procedure (Step 5 to Step 7) the rank of business processes under
  evaluation criteria is determined.
  The calculated preference indices of the treated business processes and their
  rank under the identified evaluation criteria are presented in the following text
  (Table 4, Table 5, Table 6).

Table 4 Preference indices of business processes and their rank under quality performance

| Process no. | Preference index | Rank | Degree of belief that business process can be the best |
|---|---|---|---|
| heightp=1 | (0.13,0.329,0.828) | 1 | 1 |
| p=2 | (0.086,0.211,0.559) | 3 | 0.784 |
| p=3 | (0.085,0.114,0.294) | 4 | 0.432 |
| p=4 | (0.041,0.092,0.226) | 5 | 0.288 |
| p=5 | (0.097,0.247,0.379) | 2 | 0.752 |
| height | | | |

Table 5 Preference indices of business processes and their rank under environmental protection performance

| Process no. | Preference index | Rank | Degree of belief that business process can be the best |
|---|---|---|---|
| heightp=1 | (0.065,0.172,0.5) | 3 | 0.715 |
| p=2 | (0.111,0.327,0.915) | 1 | 1 |
| p=3 | (0.099,0.298,0.801) | 2 | 0.959 |
| p=4 | (0.031,0.083,0.248) | 5 | 0.359 |
| p=5 | (0.041,0.118,0.339) | 4 | 0.522 |
| height | | | |

Table 6 Preference indices of business processes and their rank under safety performance

| Process no. | Preference index | Rank | Degree of belief that business process can be the best |
|---|---|---|---|
| heightp=1 | (0.087,0.166,0.447) | 4 | 0.721 |
| p=2 | (0.106,0.298,0.68) | 1 | 1 |
| p=3 | (0.072,0.181,0.494) | 3 | 0.768 |
| p=4 | (0.049,0.129,0.328) | 5 | 0.568 |
| p=5 | (0.089,0.225,0.578) | 2 | 0.866 |
| height | | | |

Table 7 The overall preference index

| Process no. | Preference index | Rank | Degree of belief that business process can be the best |
|---|---|---|---|
| heightp=1 | (0.065,0.231,0.9) | 2 | 0.956 |
| p=2 | (0.066,0.269,1) | 1 | 1 |
| p=3 | (0.067,0.175,0.697) | 4 | 0.869 |
| p=4 | (0.028,0.107,0.409) | 5 | 0.677 |
| p=5 height | (0.055,0.215,1.686) | 3 | 0.918 |

***B4 Missing originality.***

*The specificity of the claimed "modified" FAHP (MFAHP) method proposed by the authors is not demonstrated nor stated. The core of the proposed algorithm (steps # 5-8 of Sect.2.1) is just a few standard rules taken from the literature, while the rest (steps # 1-4 of Sect.2.1) is just definitions. Unless the authors clearly state where the originality of the proposed algorithm is, I think they cannot claim to have developed a new method: they just made an application of an existing one, and the use of the dedicated acronym MFAHP is not justified, in my opinion.*

**Response:**

Authors have started from the work of Chang (1996). In the literature, there is a wide range of variations of this work (like our manuscript). For example, the calculation of weights or preferences may be performed in different ways (Torfi et al., 2010).
F. Torfi, R.Z. Farahani, S. Rezapour, Fuzzy AHP to determine the relative weights of evaluation criteria and Fuzzy TOPSIS to rank the alternatives. Applied Soft Computing,

10 (2) (2010), 520-528.

We agree with the reviewer that proposed fuzzy AHP is not significantly modi- fied so term modified has been deleted in the revised manuscript.

**3. The model for evaluation of seaport business processes**

The proposed evaluation procedure can be realized in a way that is presented in fig. 1.

**Fig. 1 The evaluation procedure of seaport business processes by AHP**

(Please see the Fig. 3 below)

**_B5 Not fully justified conclusions._**

_According to Tab.1 and Tab.3 the "business activities in seaport" process (p = 5) gets rank 1 for both the quality and the safety performance. How can a business activity be the most crucial action for enhancing safety of a harbour? The authors comment this surprising finding by stating that "the level of customers' satisfaction mostly depends on quality of this business process realisation, so the obtained result is expected" (P15, rows 23-24). I actually thought that the focus of the paper was to establish priori- ties for the port management without a specific perspective on customers, but in view of multi-criteria optimization. If instead the authors mean that the whole analysis is just functional to enhance customers' satisfaction, then the title, abstract and scope of the paper should be consequently restricted. In any case, I cannot easily accept that business activities will enhance safety of a harbour. I think that either there is some numerical manipulation mistake or the initial expert assessments (including their rela- tive weights) were biased. This leads me back to the observation about expert weights_

*(B2) and missing analysis of consistency of the pairwise comparison matrices (B3).*

**Response:**

Authors want to thank the reviewer for this suggestion. In interaction with the expert team, we have obtained improved input data, so new tables with results are presented (table 4, table 5, table 6 and table 7; figure 2 and figure 3).
According to the final score, the business process (p=2) is the most preferred because it has the highest priority. According to the calculated degree of belief, it may be assumed that all identified processes are significant for the seaport so, in the same time, it can be suggested that the management team has defined an adequate reference model of an organization.

**Anonymous Reviewer: C-Technical comments**

*C1*

*For a symmetry reason, on P4 it seems to me much more natural to define VL=(1,2,2) and not VL=(1,1,2): just plot the 5 fuzzy numbers VL, L, M, H, VH and see why. Actually it would help the reader in having this plot as a Figure of the manuscript.*

**Response:**

Authors have used 5 linguistic expressions which are modelled by using TFNs. The domains of these fuzzy numbers are defined on the set of real line into 1-5. As there are no formal guidelines and rules to determine granulation of TFNs, authors assumed than modal values of employed TFNs should be denoted as 1, 2, 3, 4, 5, respectively.

*C2*

*Matrix on P10, row 10 (please, use symbols for identifying mathematical objects more easily!):*

- *I guess the $\times 3$ matrix refers to the K = 3 performances and each fuzzy number in the 4-tuples refers to an expert judgement. If this is correct, it should be clearly stated. Furthermore, for consistency of notation, the diagonal elements should be 4-tuples of crisp numbers, something like (1,1,1),(1,1,1),(1,1,1),(1,1,1) that could be conveniently replaced by a convenient multi-dimensional identity symbol such as the one expressed in LaTeX by mathbb1.*

- *In the following, (P12-15), also $\times 4$ (P11) and $\times 5$ appear. It would be good to always state what this dimensionality refers to. I suppose that they refer to $J_k = 4$ KPIs of each performance, and to the I = 5 "business processes", see Tab. 1 of this review.*

**Response:**

For the reason of symmetry, the elements on the main diagonal are changed in compliance with the reviewer suggestion. In the same time, the dimension of matrices are denoted.

*C3*

- *on P4, row21: replace "consensus" by "group consensus" and make reference to Step 5 (P5) of the algorithm.*

- *it is unecessary to define again* $\widetilde{W}^e_{kk'} = (...)$ *and* $\widetilde{W}^e_{jj'} = (...)$, *on P6, row 10, after they were introduced in Sect2.2*

- *Remove range of indexes (i=...j=..k=..) in both Eq.(4) and Eq.(5): they were already introduced in Sect2.2;*

- *Eq.(5) could be better rewritten as:*

$$\widetilde{a}_i = \sum_{k=1}^{K} \widetilde{w}_k \widetilde{a}_i^k = \sum_{k=1}^{K} \sum_{j=1}^{J_k} p_{ij}^k$$

- *There is a logical need to insert a separation (new subsection) on P10, row8.*

**Response:**

- Authors have changed the text in compliance with the reviewer's suggestion. (They make a decision by group consensus.)

- Authors have changed the text in compliance with the reviewer's suggestion.

- Authors have changed the text in compliance with the reviewer's suggestion.

- Authors have changed the text in compliance with the reviewer's suggestion.
  $\widetilde{a}_i = \sum_{k=1}^{K} \widetilde{w}_k \cdot \widetilde{a}_i^k$, i=1..I, j=1..$J_k$, k=1..K.          Eq. (5)

- New subsection has been incorporated as reviewer suggested.
  4.1 Business processes' ranking on real life data

---

## Author Comment (AC3) · 27 Jul 2016

Dear Ms Archetti,

I would like to inform you that authors have posted on the open discussion the Responses to reviewer's comments, according to your kind suggestion. We remain at your disposal for any further information.

Respectfully,

on behalf of the authors,

Pavle Popovic
* * *
[Figure]

2016.

---

## Author Comment (AC4) · 8 Aug 2016

Please find enclosed, as separate *.pdf files in the compressed supplement, the document with the authors' response to the Reviewer's comments, as well as the revised manuscript. We remain at your disposal for any further information.

Respectfully, on behalf of all authors,

Pavle Popovic

Please also note the supplement to this comment:
http://www.nat-hazards-earth-syst-sci-discuss.net/nhess-2016-126/nhess-2016-126-AC4-supplement.zip

---

## Short Comment (SC1) · 10 Aug 2016

I have to admit I found a bit cumbersome the reading of the paper, in the current form.

In particular, I think too many technicalities are included in the main part of the paper, making complex the reading.

I would suggest the authors to restructure section 2.3 (algorithm) and section 4 (application) to make the paper more easily readable. I.e., one option would be to clarify the main steps (of the algorithm) and results (of the application) in the main part of the paper, moving the more technical issues to an appendix of the paper itself.

In the current version, I find difficult to judge the quality of the presented work.

[Figure]

2016.

---

## Author Comment (AC5) · 22 Aug 2016

**Response to the Reviewer's comment**

**Reviewer's comment**

*I have to admit I found a bit cumbersome the reading of the paper, in the current form.*

*In particular, I think too many technicalities are included in the main part of the paper, making complex the reading.*

*I would suggest the authors to restructure section 2.3 (algorithm) and section 4 (application) to make the paper more easily readable. I.e., one option would be to clarify the main steps (of the algorithm) and results (of the application) in the main part of the paper, moving the more technical issues to an appendix of the paper itself.*

*In the current version, I find difficult to judge the quality of the presented work.*

**Response:**

We want to thank for the useful suggestions.

Authors have carefully analyzed the suggestions of reviewer. In that manner, authors have put significant effort to incorporate all suggestions into revised manuscript. We believe that the overall quality of the manuscript has been improved.

Appendix has been added to the revised manuscript. It contains detailed calculation so only main results are presented in the revised manuscript. Sections related to presentation of the model and algorithm have been merged to single section in compliance to the reviewer's suggestions.

Please find enclosed, as a *pdf. file in the supplement, the revised manuscript.

We remain at your disposal for any further information.

Please also note the supplement to this comment:
http://www.nat-hazards-earth-syst-sci-discuss.net/nhess-2016-126/nhess-2016-126-AC5-supplement.pdf

———————————————

[Figure]

**Supplement:**

**The evaluation and enhancement of quality, environmental protection and seaport safety by using fuzzy AHP**

Danijela Tadic1, Aleksandar Aleksic1, Pavle Popovic2, Slavko Arsovski1, Ana Castelli3, Danijela Joksimovic3, Miladin Stefanovic1

[revised manuscript text omitted]

**3. The model for evaluation of seaport business processes**

The proposed evaluation procedure can be realized in a way that is presented in fig. 1.

Figure 1. The evaluation procedure of seaport business processes by FAHP

30 The evaluation procedure should be delivered by the expert team which is consisted of the seaport owner, main manager, local government and the operational management of the seaport. Formally, this expert team is presented by a set of indices {1, ..., e, ..., E}. The index for an expert is denoted as e, and E is the total number of experts. The members of the expert team have different influence in the considered decision making process. The importance of experts,  $w_e$ , e=1,..,E should be determined with respect to the results of good practice.

The identified performances can be presented by the set of indices  $\{1, ..., k, ..., K\}$ . The index for a performance is denoted as k, k=1,...,K and K is the total number of identified performances. Each performance k, k=1,...,K is decomposed into KPIs. Generally, KPIs under performance k, k=1,...,K are presented by the set of indices  $\{1, ..., j, ..., J_k\}$ .

- Experts and operational managers use the pre-defined linguistic expressions, which are modelled by triangular fuzzy numbers (TFNs). The shape of the membership functions can be obtained based on one's experience, the subjective belief of decision makers, and their knowledge. Jointly used shapes of triangular function offer a good compromise between descriptive power and computational simplicity.
- 10 The total number of KPIs under performance k, k=1,..,K is denoted as  $J_k$ , and j is the index for KPI j,  $j=1,..,J_k$ .

The fuzzy rating of the relative importance of each pair of performances and their KPIs are described by each expert and presented by TFN  $\tilde{W}_{kk'}^e = (x; l_{kk'}^e, m_{kk'}^e, u_{kk'}^e)$ , k=1,...,K, and  $\tilde{V}_{jj'}^e = (x; l_{jj'}^e, m_{jj'}^e, u_{jj'}^e)$ , j=1,...,*Jk*.

The aggregation of individual opinions into a group consensus is calculated by the performed Fuzzy Ordered Weighted Aggregation (FOWA) operator (Merigo and Casanovas, 2008). The aggregated value of the considered variables are (Eq. (1)):

$$\widetilde{W}_{kk'} = (x; l_{kk'}, m_{kk'} u_{kk'}) = \sum_{e=1}^{E} w_e \cdot \widetilde{W}_{kk'}^e, k = 1, \dots, K; e = 1, \dots, E$$
Eq. (1)

Similarly, the aggregated value of the relative importance of each pair of KPIs under the identified performance is determined.

Fuzzy pair-wise comparison matrices of the relative importance of performance, the relative importance of KPIs under each performance and preference of business processes respecting each KPI are stated. It is necessary to determine the coefficient

- of consistency to reflect the consistency of the decision makers' judgements during the evaluation phase by using eigen vector method (Saaty, 2008). The eigen vector method represents a natural measure for inconsistency and it is used in wide literature and it is used in this paper, too. It is worth to mention that all relevant indexes of consistence (C.I.) should be equal or below the threshold of 0.1. The weights vector of performances and weights vector of KPIs under each performance and
- 25 the preference vector of business processes with respect to each KPI are determined by FAHP which is developed in (Wu et al., 2004).

The developed procedure is illustrated on the example of determination of the performances' weights vector in compliance with Eq. (2) and Eq. (3).

Order

5

15

30
$$\alpha_{k} = \left[\prod_{k=1}^{K} l_{kk'}\right]^{1/K}, \quad \beta_{k} = \left[\prod_{k=1}^{K} m_{kk'}\right]^{1/K}, \quad \text{and} \quad \gamma_{k} = \left[\prod_{k=1}^{K} u_{kk'}\right]^{1/K}, \quad k = 1, ..., K$$
 Eq. (2)

and

$$\alpha = \sum_{k=1}^{K} \alpha_k, \quad \beta = \sum_{k=1}^{K} \beta_k \text{ and } \gamma_k = \sum_{k=1}^{K} \chi_k, \quad k = 1, \dots, K$$

Then the weight of performance, k=1,...,K is calculated as:

$$W_k = (\alpha_k \gamma^{-1}, \beta_k \beta^{-1}, \gamma_k \alpha^{-1}) = (l_k, m_k, u_k)$$
 Eq. (3)

In a similar way (Eq. (2) and Eq. (3)), the weight of KPI j,  $\tilde{v}_{j}^{k} = (y; l_{j}^{k}, m_{j}^{k}, u_{j}^{k})$ , j=1,...,Jk; k = 1,...,K and preference of

5 business process i  $\stackrel{\sim}{p}_{ij}^{k} = (y; l_i^j, m_i^j, u_i^j)$ , i=1,..,I are calculated.

The reference model of an organization (in this case a seaport) may be seen as a general model which can be used for gaining other forms of models (Spiegel and Caulliraux, 2012). In compliance with this, an organization may be viewed as a network of interrelated processes that are focused towards achieving organizational goals (Oakland, 2004). The defining of seaport business processes is based on the process approach (ISO 9000:2008), and assessment of seaport operational management

(quality manager, environmental manager and security manager). The identified business processes are presented by the set of indices {1, ..., i, ..., I}. The total number of treated business processes is I and i, i=1,...,I is the index of the business process. The assessment of the relative preference value of each pair of business processes is achieved by group consensus. The ranking of business processes is performed according to the overall index of preference. The preference index of business process i, i=1,...,I under performance k can be calculated as (Eq. (4)):

15
$$\tilde{a}_{i}^{k} = \sum_{j=1}^{J_{k}} \tilde{v}_{j}^{k} \cdot \tilde{p}_{ij}^{k}, i=1,...,I; j=1,...,J_{k}; k = 1,...,K$$
 Eq. (4)

The overall preference index of each business process is described by a TFN.

The overall preference index of business process i, i=1,..,I can be calculated as (Eq. (5)):

$$\tilde{a}_{i} = \sum_{k=1}^{K} \tilde{w}_{k} \cdot \tilde{a} \quad \stackrel{k}{i} i=1,...,I; j=1,...,J_{k}; k = 1,...,K$$
 Eq. (5)

The rank of business processes corresponds to the rank of TFNs which are described by overall indices' preferences.

20 The ranking of the TFNs  $\tilde{p}_i$ , i=1,..,I and the calculating of the degree of belief that other business processes can be better than the business process which is placed in first place in the rank are based on a method for comparison of fuzzy numbers (Bass and Kwakernaak, 1977; Dubois and Prade, 1979).

**4 Application of FAHP in business processes' ranking**

The proposed model is tested on Kotor seaport located in a region which is protected under national legislation. In recent

25 years, the seaport has been certified with ISO 9001:2008 and ISO 14001:2004. This seaport is a relatively small port so this fact is taken into account during the definition of a reference model of the organization.

In literature from business process management, processes of seaport services represent the processes of realization (Arsovski, 2013). The number and type of business processes in a seaport is defined with respect to American Productivity and Quality Center (APQC) Process Classification Framework (PCF) and process owner opinion (Table 1). A short description of the selected business processes in ports is further discussed.

5

**Table 1 Identified business processes in the seaport**

*Planning and service monitoring* (p=1). It covers a set of activities to be implemented under the common goal of the process (responsibility for each activity, resources, timelines, and desired outputs from each activity in terms of the characteristics of

10 services and processes). This process corresponds to the process Plan for and align supply chain resources which is defined in APQC specification.

*Technology management of service providing* (p=2). It covers standard procedures for access of the vessels to the port, vessel pilotage procedures, maintenance procedures of vessels, port transportation, disembarking procedures, and procedures for cleaning, etc.

15 *Maintenance of infrastructure* (p=3). It covers maintenance procedures of docks, cranes, as well as other transport manipulating systems, warehouses, roads, etc. This process corresponds to the process of Manage Logistics and Warehousing (respecting APQC).

*Management of Environmental Health and Safety* (p=4). It is defined in compliance with APQC specification and it is important from the perspective of seaport sustainability. The effectiveness of this business process is important for the management of the port and the local and state administration.

*Activities in seaport* (p=5). This is a complex business process where a lot of different activities are defined and realized according to APQC and literature data (Medison, 2005). These activities are: material purchase, service delivery to seaport customers, marketing and service sale, management of customer demands, management of information technology and knowledge, management of financial resources and management of external relations.

25

20

**4.1 Business processes' ranking on real life data**

Used notation is provided in table 2.

**30 Table 2 Notation**

Based on the internal policy of treated seaport, the expert team is adjoined with different specific weights (table 3).

Table 3 Specific weights of expert team

The elements of constructed fuzzy pair-wise matrices are defuzzified, and after that, the consistence of fuzzy pair-wise matrices is determined. It is determined by analogy with Torfi et al., (2010).

5 Then the weight of quality performance (k=1) is calculated in compliance with Eq. (2) and Eq. (3):

$$\tilde{w}_1 = (0.89 \cdot 3.95^{-1}, 1.25 \cdot 3.22^{-1}, 1.53 \cdot 2.65^{-1}) = (0.24, 0.39, 0.58)$$

Similarly, weights of the rest of the performances are calculated:

 $\tilde{w}_2 = (0.12, 0.17, 0.25)$ , and  $\tilde{w}_3 = (0.30, 0.44, 0.66)$ .

The weights of sub criteria under quality performance are:

10
$$v_1^{-1} = (0.19, 0.28, 0.38), \quad v_2^{-1} = (0.19, 0.29, 0.41), \quad v_3^{-1} = (0.09, 0.14, 0.25), \text{ and } \quad v_4^{-1} = (0.21, 0.28, 0.41).$$

The weights of KPIs under environmental protection performance are:

$$\tilde{v}_1^2 = (0.09, 0.14, 0.25), \ \tilde{v}_2^2 = (0.21, 0.39, 0.59), \ \tilde{v}_3^2 = (0.11, 0.20, 0.36), \text{ and } \ \tilde{v}_4^2 = (0.18, 0.27, 0.48)$$

The weights of KPIs under safety performance are:

$$\tilde{v}_1^3 = (0.26, 0.38, 0.52), \ \tilde{v}_2^3 = (0.29, 0.41, 0.58), \ \tilde{v}_3^3 = (0.09, 0.14, 0.22), \text{ and } \ \tilde{v}_4^2 = (0.06, 0.07, 0.11).$$

15 The preference of KPIs under each considered performance are presented.

Quality performance

(Q1) Quality of the seaport services

$$\tilde{p}_{11}^{-1} = (0.12, 0.22, 0.41)$$
  $\tilde{p}_{21}^{-1} = (0.11, 0.19, 0.38), \tilde{p}_{31}^{-1} = (0.05, 0.07, 0.14), \tilde{p}_{41}^{-1} = (0.05, 0.08, 0.14)$  and  $\tilde{p}_{51}^{-1} = (0.25, 0.43, 0.68).$

(Q2) Average number of customers

20
$$p_{12}^{-1} = (0.28, 0.43, 0.62), p_{22}^{-1} = (0.14, 0.19, 0.29), p_{32}^{-1} = (0.04, 0.06, 0.09), p_{42}^{-1} = (0.07, 0.10, 0.16) \text{ and } p_{52}^{-1} = (0.14, 0.23, 0.34)$$

(Q3) Average number of vessels in the queue

(Q4) Pilotage operation of the vessel

$$\stackrel{\sim}{p_{14}}^{1} = (0.15, 0.29, 0.58), \quad \stackrel{\sim}{p_{24}}^{1} = (0.14, 0.28, 0.54), \quad \stackrel{\sim}{p_{34}}^{1} = (0.09, 0.21, 0.34), \quad \stackrel{\sim}{p_{44}}^{1} = (0.06, 0.1, 0.14) \text{ and } \stackrel{\sim}{p_{54}}^{1} = (0.07, 0.13, 0.26)$$

25

Environmental protection

(E1) Quality of air

$$\tilde{p}_{11}^2 = (0.11, 0.18, 0.32), \tilde{p}_{21}^2 = (0.17, 0.27, 0.40), \tilde{p}_{31}^2 = (0.15, 0.25, 0.44), \tilde{p}_{41}^2 = (0.07, 0.12, 0.23) \text{ and } \tilde{p}_{51}^2 = (0.10, 0.17, 0.25).$$

(E2) Water quality and (E3) Noise

$$\tilde{p}_{12}^{2} = \tilde{p}_{13}^{2} = (0.09, 0.13, 0.24) , \quad \tilde{p}_{22}^{2} = \tilde{p}_{23}^{2} = (0.22, 0.34, 0.59) , \quad \tilde{p}_{32}^{2} = \tilde{p}_{33}^{2} = (0.19, 0.34, 0.51) , \quad \tilde{p}_{42}^{2} = \tilde{p}_{43}^{2} = (0.05, 0.08, 0.14) \text{ and}$$
$$\tilde{p}_{52}^{2} = \tilde{p}_{53}^{2} = (0.06, 0.11, 0.18).$$

5 (E4) Hazardous substances

 $\tilde{p}_{14}^{2} = (0.15, 0.26, 0.4), \\ \tilde{p}_{24}^{2} = (0.18, 0.33, 0.53), \\ \tilde{p}_{34}^{2} = (0.14, 0.23, 0.43), \\ \tilde{p}_{44}^{2} = (0.05, 0.07, 0.12) \text{ and } \\ \tilde{p}_{54}^{2} = (0.07, 0.11, 0.22)$

**Seaport safety**

(S1) Vessel safety

10
$$p_{11}^3 = (0.12, 0.19, 0.35), \quad p_{21}^3 = (0.14, 0.31, 0.54), \quad p_{31}^3 = (0.11, 0.19, 0.43), \quad p_{41}^3 = (0.06, 0.12, 0.23) \text{ and } \quad p_{51}^3 = (0.09, 0.19, 0.38)$$

(S2) Traffic volume

[revised manuscript text omitted]

30 process (p=2) is very sensitive to safety performance of the seaport. In that manner, KPI *Traffic volume* has greatest impact on this performance.

Business process (p=3) is the most sensitive in terms of environmental protection performance. Endangering the environment occurs during the implementation of maintenance dock activities and cranes, as well as other transport manipulating systems, warehouses, roads, etc. The most significant KPIs in the scope of this analysis are *Water quality* and *Noise*. The management initiatives that should lead to KPI values' enhancement correspond to process (p=2).

- 5 The data on figure 2 and tables from 4 to table 6, it can be concluded that the business process Management of Environmental Health and Safety (p=4) has almost equal impact on all three treated performances. Enhancement of this business process can be achieved by application of different procedures which should lead to the increase of KPI values' emphasizing safety performance. These procedures should be in compliance with international standards and directives.
- When business process (p=5) is analysed, quality performance makes the most significant impact in terms of sensitivity. The 10 most of activities generated by this process are customer oriented so low performances of this process could lead to a decrease of competitiveness and a bad image of the port. Enhancement of this process with respect to quality performance may be achieved by applying the measures for enhancement of KPI *Average number of customers*.

**5.1 Research Implications**

By comparing papers which propose a model for evaluating business processes under uncertainties, certain differences could

[revised manuscript text omitted]

**Appendix**

For the purpose of calculation, the five linguistic expressions are proposed, and they are modelled by TFNs as follows:

*very low importance/preferency*: VL = (1,1,2)5 *low importance/preferency:* L = (1,2,3)*moderate importance/preferency*: M = (2,3,4)*high importance/preferency*: H = (3,4,5)*verv high importance/preferency*: VH= (4,5,5)

10 The domains of fuzzy numbers can be defined on different scales (Ishizaka and Labib, 2009) and in this paper the domains of presented TFNs are defined into interval [1-5].

The elements of constructed fuzzy pair-wise matrices are defuzzified, and after that, the consistence of fuzzy pair-wise matrices is determined. It is determined by analogy with Torfi et al., (2010).

The fuzzy-pair wise comparison matrix of the relative importance of performances is presented (according to Step 1 of the

proposed Algorithm): 15

 $\sim$

| (1,1,1),(1,1,1),(1,1,1),(1,1,1) | M, H, (1,1,1), L                | 1/L, 1/VL, 1/L, (1,1,1)                         |
|---------------------------------|---------------------------------|-------------------------------------------------|
| 1/M, 1/H, (1,1,1), 1/L          | (1,1,1),(1,1,1),(1,1,1),(1,1,1) | 1/M, 1/H, (1,1,1), 1/VL                         |
| L,(1,1,1),L,(1,1,1)             | M, H, (1,1,1), VL               | $(1,1,1),(1,1,1),(1,1,1),(1,1,1) \rfloor_{3x3}$ |

Application of FOWA is illustrated by the following example. The aggregated relative importance of quality performance (k=1) over environmental protection performance (k=2) can be calculated as:

20
$$W_{12} = 0.4 \cdot (2,3,4) + 0.3 \cdot (3,4,5) + 0.2 \cdot (1,1,1) + 0.1 \cdot (1,2,3) = (2,2.8,3.6)$$

The fuzzy pair-wise comparison matrix of the aggregated relative importance of performances is:

$$\begin{bmatrix} (1,1,1) & (2,2.8,3.6) & (0.45,0.7,1) \\ (0.28,0.36,0.5) & (1,1,1) & (0.41,0.51,0.58) \\ (1,1.43,2.22) & (1.72,1.96,2.44) & (1,1,1) \end{bmatrix}_{3_{X3}}, C.I.=0.048$$

The procedure for calculating quality weight is presented as follows (Step 2 of the proposed Algorithm):

25
$$\alpha_1 = \left[\prod_{k=1}^3 1 \cdot 2 \cdot 0.35\right]^{1/3} = 0.89, \quad \beta_1 = \left[\prod_{k=1}^3 1 \cdot 2.8 \cdot 0.7\right]^{1/3} = 1.25, \text{ and } \chi_1 = \left[\prod_{k=1}^3 1 \cdot 3.6 \cdot 1\right]^{1/3} = 1.53$$

and

$$\alpha = \sum_{k=1}^{K} \alpha_k = 2.65, \ \beta = \sum_{k=1}^{K} \beta_k = 3.22, \text{ and } \ \chi = \sum_{k=1}^{K} \chi_k = 3.95$$

Then the weight of quality performance (k=1) is calculated in compliance with Eq. (2) and Eq. (3):

$$\tilde{w}_1 = (0.89 \cdot 3.95^{-1}, 1.25 \cdot 3.22^{-1}, 1.53 \cdot 2.65^{-1}) = (0.24, 0.39, 0.58)$$

Similarly, weights of the rest of the performances are calculated:

5  $\tilde{w}_2 = (0.12, 0.17, 0.25)$ , and  $\tilde{w}_3 = (0.30, 0.44, 0.66)$ .

The fuzzy pair wise comparison matrix of the KPIs under quality performance is:

$$\begin{bmatrix} (1,1,1) & (0.78,0.95,1) & (1.25,2,2.70) & (0.85,1,1) \\ (1,1.05,1.28) & (1,1,1) & (1.4,2.10,2.90) & (0.5,1,1) \\ (0.37,050,0.80) & (0.34,0.48,0.71) & (1,1,1) & (0.34,0.55,0.87) \\ (1,1,1.18) & (1,1,2) & (1.15,1.82,2.94) & (1,1,1) \end{bmatrix}_{4_{x4}}, C.I.=0.1$$

10 By using the procedure developed in (Wu et al., 2004), the weights of sub criteria under quality performance are:

$$\tilde{v}_1 = (0.19, 0.28, 0.38), \ \tilde{v}_2 = (0.19, 0.29, 0.41), \ \tilde{v}_3 = (0.09, 0.14, 0.25), \text{ and } \ \tilde{v}_4 = (0.21, 0.28, 0.41).$$

The fuzzy pair wise comparison matrix of the KPIs under environmental protection performance is:

| (1,1,1)          | (0.22,0.25,0.40) | (0.31,0.48,0.80) | (1,1,1.30)       |             |
|------------------|------------------|------------------|------------------|-------------|
| (2.50,3.57,4.55) | (1,1,1)          | (1.40,2.10,2.90) | (0.50,1,1)       | . C.I.=0.91 |
| (1.25,2.08,3.23) | (0.34,0.48,0.71) | (1,1,1)          | (0.34,0.55,0.85) | ,           |
| (0.87,1,1)       | (1,1,2)          | (1.18,1.82,2.94) | (1,1,1)          | 4x4         |

15 The weights of KPIs under environmental protection performance are:

$$\tilde{v}_{1}^{2} = (0.09, 0.14, 0.25), \ \tilde{v}_{2}^{2} = (0.21, 0.39, 0.59), \ \tilde{v}_{3}^{2} = (0.11, 0.20, 0.36), \text{ and } \tilde{v}_{4}^{2} = (0.18, 0.27, 0.48)$$

The fuzzy pair wise comparison matrix of the KPIs under safety criterion is:

20 The weights of KPIs under safety performance are:

$$v_1^{3} = (0.26, 0.38, 0.52), v_2^{3} = (0.29, 0.41, 0.58), v_3^{3} = (0.09, 0.14, 0.22), \text{ and } v_4^{2} = (0.06, 0.07, 0.11)$$

Similarly, the fuzzy pair-wise comparison matrices of the business processes' preference are presented.

**Ouality performance**

(Q1) Quality of the seaport services

| (1,1,1) | L       | L       | H       | 1/M     |              |
|---------|---------|---------|---------|---------|--------------|
| 1/L     | (1,1,1) | M       | M       | 1/L     |              |
| 1/L     | 1/M     | (1,1,1) | 1/L     | 1/VH    | , C.I.=0.058 |
| 1/H     | 1/M     | L       | (1,1,1) | 1/VH    |              |
| Μ       | L       | VH      | VH      | (1,1,1) | 5x5          |

5  $\stackrel{\sim}{p_{11}} = (0.12, 0.22, 0.41) \stackrel{\sim}{p_{21}} = (0.11, 0.19, 0.38), \stackrel{\sim}{p_{31}} = (0.05, 0.07, 0.14), \stackrel{\sim}{p_{41}} = (0.05, 0.08, 0.14) \text{ and } \stackrel{\sim}{p_{51}} = (0.25, 0.43, 0.68).$

(Q2) Average number of customers

(1,1,1) VH Η MΗ 1/*VH* (1,1,1) *VL* VL1/H1/H 1/VL (1,1,1) 1/H 1/VH, C.I.=0.085 1/M 1/VL H (1,1,1) 1/LVH $L = (1,1,1) \Big|_{5=5}$ 1/HΗ

$$\tilde{p}_{12}^{-1} = (0.28, 0.43, 0.62), \tilde{p}_{22}^{-1} = (0.14, 0.19, 0.29), \tilde{p}_{32}^{-1} = (0.04, 0.06, 0.09), \tilde{p}_{42}^{-1} = (0.07, 0.10, 0.16) \text{ and } \tilde{p}_{52}^{-1} = (0.14, 0.23, 0.34)$$

10 (Q3) Average number of vessels in the queue

$$\begin{bmatrix} (1,1,1) & VH & H & L & M \\ 1/VH & (1,1,1) & L & L & VL \\ 1/H & 1/L & (1,1,1) & L & VL \\ 1/L & 1/L & 1/L & (1,1,1) & 1/M \\ 1/M & 1/VL & 1/VL & M & (1,1,1) \end{bmatrix}_{5x5} , C.I.=0.093$$

$$\overset{1}{p_{13}} = (0.25, 0.44, 0.72), \ \overset{1}{p_{23}} = (0.1, 0.17, 0.3), \ \overset{1}{p_{33}} = (0.08, 0.13, 0.26), \ \overset{1}{p_{43}} = (0.06, 0.09, 0.2) \text{ and } \overset{1}{p_{53}} = (0.12, 0.17, 0.27).$$

(Q4) Pilotage operation of the vessel  $\lceil (1,1,1) \quad L \quad L \quad VL \quad 1/M \rceil$

15

M

 $\tilde{p}_{14}^{1} = (0.15, 0.29, 0.58), \tilde{p}_{24}^{1} = (0.14, 0.28, 0.54), \tilde{p}_{34}^{1} = (0.09, 0.21, 0.34), \tilde{p}_{44}^{1} = (0.06, 0.1, 0.14) \text{ and } \tilde{p}_{54}^{1} = (0.07, 0.13, 0.26)$ Environmental protection

(E1) Quality of air

 $\begin{bmatrix} (1,1,1) & 1/L & 1/L & M & VL \\ L & (1,1,1) & VL & M & VL \\ L & 1/VL & (1,1,1) & L & VL \\ 1/M & 1/M & 1/L & (1,1,1) & L \\ 1/VL & 1/VL & 1/VL & 1/L & (1,1,1) \end{bmatrix}_{5x5},$   $\tilde{p}_{11}^{2} = (0.11, 0.18, 0.32), \tilde{p}_{21}^{2} = (0.17, 0.27, 0.40), \tilde{p}_{31}^{2} = (0.15, 0.25, 0.44), \tilde{p}_{41}^{2} = (0.07, 0.12, 0.23) \text{ and } \tilde{p}_{51}^{2} = (0.10, 0.17, 0.25).$

5 (E2) Water quality and (E3) Noise

 $\begin{bmatrix} (1,1,1) & 1/M & 1/M & H & VL \\ M & (1,1,1) & VL & H & H \\ M & 1/VL & (1,1,1) & H & H \\ 1/H & 1/H & 1/H & (1,1,1) & L \\ 1/VL & 1/H & 1/L & 1/L & (1,1,1) \end{bmatrix}_{5x5}, C.I.=0.77$

$$\tilde{p}_{12}^{2} = \tilde{p}_{13}^{2} = (0.09, 0.13, 0.24) , \quad \tilde{p}_{22}^{2} = \tilde{p}_{23}^{2} = (0.22, 0.34, 0.59) , \quad \tilde{p}_{32}^{2} = \tilde{p}_{33}^{2} = (0.19, 0.34, 0.51) , \quad \tilde{p}_{42}^{2} = \tilde{p}_{43}^{2} = (0.05, 0.08, 0.14) \text{ and } \tilde{p}_{52}^{2} = \tilde{p}_{53}^{2} = (0.06, 0.11, 0.18).$$

10

(E4) Hazardous substances

 $\begin{bmatrix} (1,1,1) & 1/VL & 1/VL & VH & M \\ VL & (1,1,1) & VL & H & M \\ VL & 1/VL & (1,1,1) & VH & L \\ 1/VH & 1/H & 1/VH & (1,1,1) & 1/M \\ 1/M & 1/M & 1/L & M & (1,1,1) \end{bmatrix}_{5x5}^{2}, \text{ C.I.=0.016}$

15 Seaport safety

(S1) Vessel safety

 $\begin{bmatrix} (1,1,1) & VL & 1/VL & VL & VL \\ 1/VL & (1,1,1) & L & M & L \\ VL & 1/L & (1,1,1) & L & VL \\ 1/VL & 1/M & 1/L & 1/VL & 1/L \\ 1/VL & 1/L & VL & L & (1,1,1) \end{bmatrix}_{5x5} , C.I.=0.03$

$$\tilde{p}_{11}^3 = (0.12, 0.19, 0.35), \quad \tilde{p}_{21}^3 = (0.14, 0.31, 0.54), \quad \tilde{p}_{31}^3 = (0.11, 0.19, 0.43), \quad \tilde{p}_{41}^3 = (0.06, 0.12, 0.23) \text{ and } \quad \tilde{p}_{51}^3 = (0.09, 0.19, 0.38).$$

(S2) Traffic volume

| [(1,1,1) | VL      | (1,1,1) | VL      | VL      |              |
|----------|---------|---------|---------|---------|--------------|
| 1/VL     | (1,1,1) | M       | H       | M       |              |
| (1,1,1)  | 1/M     | (1,1,1) | M       | L       | , C.I.=0.069 |
| 1/VL     | 1/H     | 1/M     | (1,1,1) | 1/VL    |              |
| 1/VL     | 1/M     | 1/L     | VL      | (1,1,1) | 5x5          |

5  $p_{12}^{3} = (0.14, 0.18, 0.36), \quad p_{22}^{3} = (0.2, 0.37, 0.56), \quad p_{32}^{3} = (0.12, 0.21, 0.35), \quad p_{42}^{3} = (0.06, 0.11, 0.16) \text{ and } \quad p_{52}^{3} = (0.08, 0.13, 0.24)$

(S3) Weather sea condition and channel condition

$$\begin{bmatrix} (1,1,1) & VL & 1/L & 1/VH \\ 1/VL & (1,1,1) & VL & L & 1/H \\ L & 1/VL & (1,1,1) & 1/M & 1/H \\ L & 1/L & M & (1,1,1) & 1/M \\ VH & H & H & M & (1,1,1) \end{bmatrix}_{5x5}^{,3}, C.I.=0.084$$

10

(S4) Other safety factors

$$\begin{bmatrix} (1,1,1) & VL & 1/L & 1/VL & 1/VH \\ 1/VL & (1,1,1) & L & 1/M & 1/H \\ L & 1/L & (1,1,1) & 1/M & 1/M \\ VL & M & M & (1,1,1) & 1/L \\ VH & H & M & L & (1,1,1) \end{bmatrix}_{5x5},$$
, C.I.=0.088
$$\tilde{p}_{14}^{3} = (0.07, 0.12, 0.19), \quad \tilde{p}_{24}^{3} = (0.06, 0.12, 0.20), \quad \tilde{p}_{34}^{3} = (0.06, 0.11, 0.25), \quad \tilde{p}_{44}^{3} = (0.13, 0.23, 0.45) \text{ and } \tilde{p}_{54}^{3} = (0.24, 0.44, 0.71)$$

**15 References**

[revised manuscript text omitted]

---

## Author Comment (AC6) · 11 Sep 2016

Please find enclosed, as separate *.pdf files in the compressed supplement, the document with the authors' final response to the reviewers' comments, as well as the revised manuscript.

We remain at your disposal for any further information.

Respectfully, D.Tadic, A. Aleksic, P.Popovic, S. Arsovski, A. Castelli, D. Joksimovic and M. Stefanovic

Please also note the supplement to this comment:
http://www.nat-hazards-earth-syst-sci-discuss.net/nhess-2016-126/nhess-2016-126-AC6-supplement.zip